# A comparison of several media types and basic techniques used to assess outdoor airborne fungi in Melbourne, Australia

**Wesley D. Black** ⓘ *

Biotopia Environmental Assessment, Melbourne, Victoria, Australia

* wesblack@biotopia.net.au

## Abstract

Despite the recent increase in interest in indoor air quality regarding mould, there is no universally accepted standard media for the detection of airborne fungi, nor verification of many commonly used techniques. Commonly used media including malt-extract agar (MEA), Sabouraud dextrose agar (Sab), potato dextrose agar (PDA) with and without antibiotics chloramphenicol & gentamycin (CG) were compared for their suitability in detecting a range of airborne fungi by collecting 150 L outdoor air on a number of different days and seasons via an Anderson 400-hole sampler in suburban Melbourne, Australia. There was relatively little variation in mean numbers of colony forming units (CFU) and types of fungi recovered between MEA, PDA, Sab media groups relative to variation within each group. There was a significant difference between Sab, Dichloran-18% glycerol (DG18) and V8® Original juice agar media, however. Antibiotics reliably prevented the growth of bacteria that typically interfered with the growth and appearance of fungal colonies. There was no significant evidence for a growth enhancing factor from potato, mineral supplements or various vegetable juices. Differing glucose concentrations had modest effects, showing a vague ideal at 2%-4% with peptone. Sanitisation of the aluminium Andersen 400-hole sampler top-plate by flame is possible, but not strictly required nor advisable. The use of SabCG as a standard medium was generally supported.

## Introduction

Mould are a wide range of fungal organisms that flourish under damp conditions indoors and outdoors, and in humans exposure is linked to the exacerbation of asthma, allergic rhinitis and occasionally infection [1], but requires additional investigation [2].

A review of the current literature suggests there is no universally accepted method for detecting, identifying and/or enumerating fungi within buildings, and similarly a lack of universally accepted limits for maximum permissible and/or normal exposure to occupants, or even what may constitute a 'mouldy' house [3].

Outdoors, various moulds, yeasts, various other fungi and organisms saprophytically degrade organic matter such as fallen leaves, trees, etc., and are generally ecologically beneficial. The most common outdoor mould & yeast genera/types noted in studies in the Northern

**Data Availability Statement:** All relevant data are within the manuscript and its Supporting Information files.

**Funding:** Biotopia Environmental Assessment Pty Ltd provided support in the form of salary and

materials to author WDB. The specific roles of this author is articulated in the 'author contributions' section. The funder had no commercial or vested interests in study design, data collection and analysis, decision to publish, or preparation of the manuscript. No additional external funding was received for this study.

**Competing interests:** The author has read the journal's policy and has the following competing interest: WDB is the CEO and owner of Biotopia Environmental Assessment Pty Ltd (http://www. biotopia.com.au/). There are no patents, products in development or marketed products associated with this research to declare. This does not alter the author's adherence to PLOS ONE policies on sharing data and materials.

Hemisphere were *Cladosporium*, *Aspergillus*, *Penicillium*, *Alternaria*, *Candida*, *Botrytis and Helminthosporium*. Within houses not known to be problematic the most common mould & yeast genera/types noted were very similar to outdoors, but included *Epicoccum* mould and *Streptomyces* bacteria. These indoor organisms are not usually a problem except in persistently humid or wet areas of houses in which such organisms significantly grow in number. Exposures to mould varies depending on a range of factors including regional differences, local climate including outdoor humidity and wind, shade, organic debris, landscape maintenance, etc., heating and cooling systems, indoor humidity and air-filtration and ventilation systems [2]. Dampness in a house is also associated with the deterioration of structural components such as plasterboard / Gyprock / drywall panels [4].

Mould living in damp indoor environments have the potential to cause a variety of adverse health effects by several mechanisms leading to rhinitis, sinusitis, conjunctivitis, asthma, and actual infection [2,5].

Once a building is wet enough for fungal colonisation, remediation is required promptly to prevent further growth and also thorough physical removal of the then numerous fungal particles to reduce exposure of occupants and site workers [6,7]. There are significant differences of opinion globally on how best to achieve this, and to what degree, and how to objectively determine if it has actually been achieved [2,8–14]. This is curious given the increasing number of legal disputes at least in the state of Victoria, Australia [15] and the Australian Federal Government interest [16].

The main established means of determining if a building is mouldy is to compare air samples taken from outdoors and in a number of locations indoors, counting fungal particles by either culture-based methods for 'viable' colony-forming units (CFU), or microscopy-based 'total-count' of identifiable fungal particles, or ideally both to overcome the limitations of each, as also airborne and settled particles, and to measure dampness and humidity [8,17].

A method of collecting airborne viable particles is the Anderson air sampler [18,19], being a calibrated air-pump drawing air through a top-plate with a set of 200 or 400 small holes directly over a Petri dish of agar gel media. Air drawn through the holes hits the gel surface, deflecting sharply and hence depositing particles on the gel. The Petri dish is then incubated to allow growth of organisms, facilitating identification and enumeration. It remains to be seen which agar media is best suited to these purposes, however.

Other methods exist such as simple swabbing and culture, or 'replicate organism detection and counting' (RODAC) touch-plates that employ a slightly raised agar gel surface that is applied to the test surface then incubated [20,21]. These, however, have been found to have a poor and variable recovery rate from standardised indicator-organism seeded surfaces between several commercial products [22]. Other problematic methods exist such as settle-plates exposed to air for a time [23–27], or the detection of adenosine triphosphate (ATP), diphosphate (ADP) and/or monophosphate (AMP) from biological substances on various surfaces but is not good for mould and bacterial spores [28,29]. LASER airborne particle counters are unable to determine if a particle is a viable spore, nonviable spore, other whole organisms, pollen, mineral grit, sawdust, skin flakes, etc. [30,31] and a study found no significant association between fungal spores and particle numbers [32]. There are no strict regulations for dusts in a house, unlike industrial clean-rooms [33].

There was a correlation between dampness, visible signs of mould and damage, and detection of viable 'indicator' moulds in buildings [34]. A study of dust extracted from carpets and rugs in many houses in Wallaceburg, Ontario, Canada, found that *Alternaria alternaria*, *Aureobasidium pullulans*, *Eurotium herbariorum*, *Epicoccum nigrum*, *Aspergillus versicolor* and *Penicillium chrysogenum* were present in 50% or more of the samples analysed [35]. A study of variously mouldy rooms in Germany used dichloran-18%-glycerol agar (DG18) and malt-

extract agar (MEA) media [34] and did draw a correlation, it remains to be seen whether these are the ideal media for these types of studies given they appear to be used more by tradition than their demonstrated suitability for use in a variety of conditions. DG18 is used as a selective media for xerotolerant and mesophilic organisms given its low water-activity ($a_w$) courtesy of its high salt/solute content [36].

DG18 has dichloran added to limit the spread of some fast-growing fungal colonies, limiting their diameter [37] and reducing the problem of covering over other, smaller colonies making identification and enumeration difficult. Yet the dichloran affects the growth of various fungi differently, barely limiting the growth of some while completely inhibiting others [38] and is somewhat toxic and hence somewhat less than ideal for handling and disposal [39].

Studies comparing various media used to sample indoor air fungi found that DG18 at 25°C generally recovered significantly higher numbers compared to MEA and/or incubation at 37°C [40]. Other workers noted that *Cladosporium halotolerans* more often survived sudden rehydration on high $a_w$ media after having been dried and cultured in low $a_w$ media than did *Aspergillus niger* and *Penicillium rubens* given these tended lyse on rehydration [41]. Others suggested that the lower $a_w$ of DG18 (~0.96) compared to other common media (~0.99) may allow better recovery of food spoilage yeasts that originally grew at low $a_w$, presumably due to osmolysis on high $a_w$ media [38]. For context, typical seawater has an $a_w$ of 0.98 [42], and many salt-preserved foods have an $a_w$ of around 0.95 [43].

Studies of dust-borne fungi in houses using media including DG18 suggested that spores may be unable to grow in culture for various reasons including inappropriate nutrients, temperature or inhibitors, and hence yielding only loose associations in numbers of mould detected in notably-mouldy and non-mouldy houses [44,45]. Given that DG18 was originally developed to enumerate xerophilic foodborne moulds and yeasts this is unsurprising [36] and it was also found that it was less able to reliably culture various food spoilage fungi compared to other newer media and hence is not now recommended even for this purpose [38]. Some moulds found in WDB such as *Stachybotrys chartarum atra* and *Chaetomium globosum* require very high $a_w$, and may lose their viability soon after collection, and are very slow growing, very often being totally obscured by faster-growing moulds like *Penicillium*, *Aspergillus*, *Ulocladium*, etc., that that tend to spread over them [46]. Thus, a low $a_w$ media such as DG18 may fail to detect important fungi.

MEA has been used for some 100 years at least, presumably because of its relative low cost and the high availability of malt extract, and presumably the likely common utilisation of the sugar, maltose, by organisms typically rotting or fermenting grains. Some brewing / baking yeast (*Saccharomyces cerevisiae*) strains have differing utilisation of maltose [47]. It is unclear if the range of various other fungi growing on skin flakes, damp cardboard and carpets, etc., in water damaged buildings (WDB) would utilise maltose given its likely absence.

A study of 64 homes in the UK used Sabouraud 4% glucose chloramphenicol agar media (SabC) did establish a correlation between visible mould and detected mould [48]. Sabouraud media was originally formulated to reliably grow a range of dermophytic fungi via the fairly universal energy source, glucose, and a general amino acid / peptide supplement, peptone, but not usually the vast numbers of resident bacteria due to an acidic pH.

Rose Bengal medium is also used [37,38], being both a selective and differential medium in that the Rose Bengal dye is taken up by fungi more than other organisms but has the problem of becoming toxic when exposed to light [49,50].

A range of other media are also occasionally used for fungi for various objectives, including the passaging/sub-culturing of reference strains that have already been isolated and purified, especially plant-borne pathogens (potato-dextrose agar, PDA) [51], food-spoilage fungi (DRBC, PCAC, TGYC, DG18) [38] or for selection/identification of microbes able to utilise

sucrose and inorganic nitrogen (Czapek-Dox Agar) [52] or enriching for specific subsets of microorganism populations (tap-water cellulose agar) [53], etc., which are hence significantly different from the objective of best estimating the numbers of viable fungi associated with WDB and normal indoor/outdoor airborne fungi. There is a shift in fungal ecology between outdoors and visibly mouldy dwellings, becoming less relatively diverse presumably because the colonising fungi is better suited to the dwelling's dampness, humidity, temperature, building materials, etc. [46,54].

The use of variously enriched media has been explored including the addition of minerals and/or vegetable juices such as the commercially available 'V8® Original' juice by the Campbell Soup Company, a blend of eight vegetables including mainly tomato juice, with the notion that there is some factor that enhances the growth and detection of fungi, especially plant pathogens [34,46,55,56]. Some supplements such as molasses, V8® juice, coconut, urea and ammonium variously affected the growth of *Aspergillus flavus* CA43 [57]. The numbers of fungi recovered from houses varied over time when using media including Rose Bengal, MEA, V8 and DG18 agar had an at least approximately 20% coefficient of variation [58].

Bacteria and fungi compete for the same resources and hence affect the growth of each other by either using limited resources faster, or actively secreting substances that inhibit their growth. Chloramphenicol (chloromycetin) is secreted by *Streptomyces venezuelae*, a soil bacterium [59]. Various *Lactobacillus* bacteria have an antifungal effect [60,61], as also *Bacillus* bacteria strains [62,63], and are included in commercial preparations of probiotic capsules promoted as of benefit to people with gastrointestinal complaints such as irritable bowel syndrome (IBS) [64].

Bacteria exist in vast numbers in ordinary topsoil, dusts, on skin, dander and hence the normal indoor environment. This is of less concern for media used to analyse typically sterile samples such as body tissues and cerebrospinal fluid. Old media formulations did not include antibiotics, often having been not stable enough to be autoclave sterilised at 121°C and stored in solution for a practical time period, and penicillin is an example of this [65,66] as also are other labile antibiotics [67] in contrast to chloramphenicol, which is far more stable and hence suitable for such use [59]. Bacteria don't cause health effects in quite the same way as moulds [68–71] and hence their exclusion from fungus-specific detection media is justifiable.

## Materials and methods

### Sabouraud / SabCG medium agar

As indicated, either pre-made complete Sabouraud Dextrose Agar powder (Oxoid CM0041, 65 g/L), mixed into cold reverse-osmosis (RO) purified water, adjusted to pH 5.6 ±0.2, autoclaved (121°C, 15 min, jacket off, no vacuum pre- or post-autoclave), cooled before adding 100x chloramphenicol-gentamycin stock (100xCG: 5 mg/mL chloramphenicol (Sigma C0378) in 50% ethanol, 40 mg/mL gentamicin (Pfizer / DBL / Pharmacia), hence 10 mL/L) before pouring. Otherwise where indicated, made 'from scratch' from individual components, being Mycological Peptone™ powder (Oxoid LP0040, 10 g/L, the powder stated by the manufacturer as having 9.5 w/w total nitrogen and 2.9% w/w amino nitrogen), glucose powder (APS/AJAX, 40 g/L), bacteriological agar powder (Oxoid LP0011, 15 g/L) mixed into cold RO water to 1 L, adjusted to pH 5.6 ±0.2, autoclaved, cooled, with or without 100xGC addition as indicated then poured.

### PDA potato dextrose agar

Pre-made complete PDA powder (Oxoid CM0139, 39 g/L, being 4 g/L potato extract, 20 g/L glucose and 15 g/L agar when poured) mixed into cold RO water to 1 L, autoclaved, cooled, with or without addition of 10 mL/L of 100xCG as indicated then poured.

### V8 media agar with chloramphenicol and gentamycin (V8c)

V8® Original vegetable juice (Campbell Soup Company, Campbell's Soup Australia, Lemnos, Victoria, 200 mL), calcium carbonate (Sigma, 2 g), bacteriological agar powder (15 g), mixed into cold RO water to 1 L, autoclaved, cooled, with or without addition of 10 mL of 100xCG as indicated then poured.

### Dichloran 18% glycerol media agar with chloramphenicol and gentamycin (DG18c)

Pre-made dichloran glycerol agar base powder (Oxoid CM0729, 31.5 g, being peptone 5 g/L, glucose 10 g/L, potassium dihydrogen phosphate 1 g/L, magnesium sulphate 0.5 g/L, dichloran 0.002 g/L and agar 15 g/L and pH 5.6 when poured), glycerol (Sigma, 176 mL) mixed into cold RO water to 1 L, autoclaved, cooled, with or without addition of 10 mL/L of 100xCG added as indicated then poured.

### Malt extract media agar without/with chloramphenicol and gentamycin (MEA, MEACG)

malt-extract powder (Oxoid LP0039, 34 g/L, the powder stated by the manufacturer as having 1.1% w/w total nitrogen, 0.6% w/w amino nitrogen, 0.1% w/w sodium chloride, and typically 60–63% w/w reducing sugars with other long-chain sugars present, and mainly soluble calcium and magnesium salts), agar powder (10 g/L), mixed into cold tap water, adjusted to pH 5.5 ±0.2, autoclaved (110˚C, 25 min), cooled, with or without 10 mL/L of 100xCG as indicated then poured.

### Peptone media agar with/without chloramphenicol and gentamycin (PeptoneCG, Peptone-only)

Mycological Peptone™ powder (Oxoid LP0040, pH 5.3 at 2%), powdered agar and other components as indicated, cold RO water, autoclaved, cooled, with or without 10 mL/L of 100xCG as indicated and then poured.

### Maltose agar media

maltose monohydrate powder (Sigma M2250, to 40 g/L when poured) was added to liquid medias as indicated prior to being autoclaved, cooled, with or without 10 mL of 100xGC addition as indicated and then poured.

### Mineral Supplement #1 (MS1) media agar

50xMS1 stock was prepared as 25 g $CaCl_2.2H_2O$ (Sigma), RO water to 100 mL, then autoclave sterilised. MS1 medium was prepared as follows: 40 g glucose, 10 g peptone, 15 g bacteriological agar, RO water to 1 L. Autoclaved, cooled to approx. 50˚C, 20 mL of 50xMS1 stock added dropwise with stirring of the liquid medium, 10 mL of 100xCG stock, pH adjusted to 6.7 +/- 0.3 and then poured.

### Mineral Supplement #2 (MS2) media agar

50xMS2 stock was prepared as 40 g Ammonium dihydrogen phosphate $(NH_4)H_2PO_4$, 5 g Potassium Chloride KCl, 5 g Magnesium Sulphate $MgSO_4.7H_2O$, 0.1 g Ferrous Sulphate $FeSO_4.7H_2O$, 0.1 g Zinc Sulphate $ZnSO_4.7H_2O$, 0.032 g Cupric Sulphate (anhydrous) $CuSO_4$, RO

Water to 100 mL then filter sterilised (Thermo Scientific™ 597–4520, 0.20 μm pore diameter). MS2 medium was prepared as per MS1 medium, but using the 50xMS2 stock.

### Vegetable supplements

Tomatoes (hydroponic 'truss' variety) and celery were bought fresh from a local supermarket (Sim's IGA Supermarket, Footscray, Victoria, Australia) and each were juiced via kitchen food processor (Sunbeam). V8® Original vegetable juice (Campbell Soup Company, in UHT sterilised bottles) were similarly acquired, being a combination of tomato, beets/beetroot, celery, carrot, lettuce, parsley, watercress, spinach juice concentrates and water.

### Clarification of vegetable juices

Coarse filtration through clean/washed calico cloth, warming filtrate to approx. 50˚C in a microwave oven, adding 1/4 volume of liquid agar stock (2% in RO water) that had been molten and cooled to approx. 50˚C prior to addition, mixed then cooled to 4˚C and filtered through clean calico cloth using some manual pressure, settled at 4˚C for 30–60 minutes in a tall jar, decanted and filter sterilised (Corning 431218, 0.20 μm pore diameter).

### Heat-treatment of vegetable juices

30 mL of freshly clarified sterile vegetable juice in a sterile 50 mL plastic tube with a slightly loose lid was placed in approx. 200 mL boiling water in a borosilicate glass vessel and placed in a microwave oven, gently boiling for approx. 10 minutes, then cooled. Then 1 mL of each sterile supplement was added to the top of previously poured, set, room-temperature and slightly dried SabCG media plates and a sterile glass L-shaped bacteriology cell-spreader (Merck, S4522) was used to spread the liquid evenly over the entire surface and allowed to soak/dry before use.

### Air sampling

QuickTake30 (SKC Biosystems) with 'SKC Biostage-400' single-stage 400-hole sampler assembly attachment [19] as per Andersen, 1958 [18] without any attachments atop the sampler top-plate, positioned horizontally [72] and at 1.5 m height, approx. 2 m from buildings, with wind-speeds approx. 1–3 m/s (4–9 km/h) on each occasion (i.e., still air and windy days were avoided, as also rain). Typically 150 L air samples were taken in triplicate in a 'collated' sequence, being all of the first plates of each different media, then all of the second plates, then all of the third plates, to best minimise the effect of differing air velocities and wind direction shifts, local dust-raising activities, etc.

### Viable CFU

The term 'viable CFU' is used to describe mould (*Actinomycetes*, *Zygomycetes*), yeasts, presumably also other fungi possibly including mushrooms, toadstools, earth-stars, puffballs, timber brown-rot, white-rot, etc., (*Basidiomycetes* generally), fungal plant pathogens and other organisms unaffected by chloramphenicol / gentamycin, slime-moulds and any other organism able to grow at 27˚C to a visually apparent size and hence detectable after 3 days on the stated media.

### Calculation of CFU/m$^3$

Was by (1.25x correction factor for the use of plastic Petri dishes as per Andersen, 1958 [18]) x (the total detected CFU/Petri dish) and then reference to the Andersen Table 1: positive hole

**Table 1. Summary of results of CFU/m³ detected in 150 L outdoor air by SabCG in various experiments performed in each season.**

| Season | Month / year | Mean CFU/m³ | Standard Deviation | Figure ID |
|---|---|---|---|---|
| Spring | September 2019 | 282 | 22 | Fig 6 |
| Summer | February 2020 | 316 | 30 | Fig 8 |
| Autumn | April 2020 | 289 | 19 | Fig 2 |
| Winter | July 2020 | 242 | 69 | Fig 4 |
| Mean | | 282 | | |

correction table [18], and then dividing the result by the number of cubic metres of air sampled (typically 0.15 m³, being 150 L of air).

## Identification of organisms

By visual (1x) inspection of colonies from upper- and under-side (reverse), and 40-100x inspection of the upper side of colonies on the medium by microscopy via Radical RXLr-3T (Radical Scientific Equipment Pvt. Ltd., India), and if required, 400-1000x by tease-mount in water, with oil immersion on top of a 0.17 mm glass cover slip at 1000x. Identification via various references [73–76] down to general categories as per ASTM D7658-17, D7391-20 [77,78].

## Flame sanitisation of Andersen 400-hole top plate

The top plate of a SKC Biostage-400 assembly was unscrewed and placed right-side-up on a bench then approx. 3–5 mL of common household methylated spirits (approx. 95% ethanol solution UN1170, with approx. 1% denaturing odorants; Digger brand, Recochem Inc., Australia) was added and lit with a butane lighter, then allowed to burn until it self-extinguished and allowed to cool and dry for 1 minute.

## Thermography

Testo model 875–2 thermographic camera with SuperResolution module active (Testo, Germany).

## Bacteria powder

'Double Strength Probiotic' powder in capsules (Life-space Probiotics company), containing a combination of *Lactobacillus rhamnosus* Lr-32 & GG & HN001, *Bifidobacterium lactis* BI-4, *L. plantarum* Lp-115, *Streptococcus thermophilus* St-21, *L. casei* Lc-11, *L. paracasei* Lpc-37, *B. animalis* ssp. *lactis* HN019, *B. breve* Bb-03, *B. longum* Bl-05, *L. gasseri* Lg-36, *B. infantis* Bi-26, *L. delbrueckii* ssp. *bulgaricus* Lb-87 and *L. reuteri* 1E1, in approximately that order by number. A single capsule of 64 billion CFU/capsule was opened and a fine stream of the powder was blown by small electric fan towards the QuickTake30 sampling unit while in operation approximately 1 m away. This was intended only as an excess of bacteria known to interfere with fungal growth at >17,520 CFU/m³ air and hence at least one CFU for each of the 400 holes of the Andersen 400-hole sampler.

## Lugol's iodine

50 mg/mL iodine in 100 mg/mL potassium iodide solution as supplied in a standard Gram staining kit (Magnacol Pty Ltd, UK).

## Glucose testing

By blood glucose test kit (Accu-Chek Mobile U1, Roche).

## Data processing and graphing

MS-Excel for Mac v16.35 (Microsoft). Analysis of variance (ANOVA) was by "Anova: Single Factor" [sic] method via Excel's analysis tools package, as also mean and standard deviation (SD).

# Results

## Media tests: Glucose, maltose, peptone and SabCG

Testing various peptone, glucose and maltose-based media (Fig 1) in late-January 2020 (summer) in Melbourne, Australia, indicated a significant variance in the number of CFU/m$^3$ detected within media types relative to their means, often greater than the variation between media types. Importantly, PeptoneCG (no sugars) had fewer CFU than SabCG 'stock' (using pre-prepared complete powder) and 'from scratch' (using the same individual components used in other media), but comparable CFU to MaltosePeptoneCG and GlucoseMaltosePeptoneCG. The AgarCG and MaltoseCG were similarly very low in detected CFU/m$^3$, and each had fungal colonies that were similarly very poorly developed and difficult to see, in no way comparable to colonies observed on the other media, and very difficult to identify although several had *Alternaria*-like chains of dark spores at the surface despite a lack of distinct hyphae,

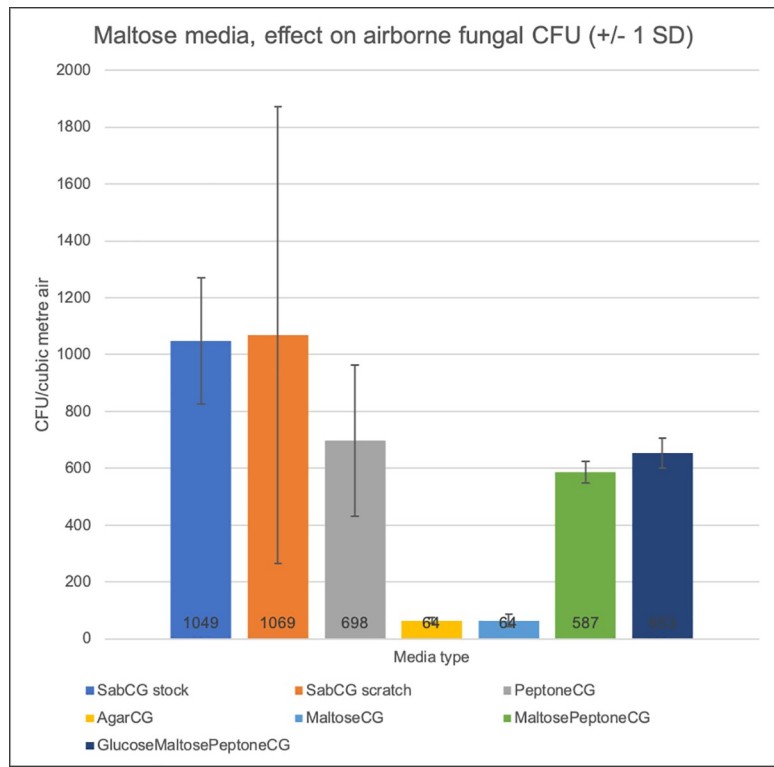

**Fig 1. Maltose, glucose and peptone utilisation by outdoor airborne fungi.** Media included SabCG 'premade stock' with CG antibiotics added, SabCG 'scratch' made from the individual components used in the other media, PeptoneCG (with no added sugars), AgarCG, 4% MaltosePeptoneCG and GlucoseMaltosePeptoneCG (hence 8% total sugars). Error bars are +/- 1 SD to better indicate the noted variance between three replicate plates of each medium.

hence definitely not the full gamut of outdoor airborne organisms, and would not normally be counted as CFU at 3 days incubation.

One-way ANOVA analysis indicated was a significant difference between all media (p = 0.0089), but for SabCG(stock), SabCG(scratch) and PeptoneCG, significantly more variation within groups than the variation between groups (p = 0.623).

The results indicated that the presence of maltose reduced the numbers of CFU detected, being similar to PeptoneCG despite the noted degree of hydrolysis of maltose to glucose. The variation was quite significant and hence the association was not very strong at the number of replicate plates used per experiment, however.

That the PeptoneCG medium showed appreciable numbers of CFU was somewhat surprising as the standard understanding is that fungi typically require carbohydrates. The standard SabCG with 2–4% glucose seems OK for practical purposes, but the 4% glucose (the standard formula of many years) is likely slightly more useful in comparisons with historical data.

That the GlucoseMaltosePeptoneCG was comparable to MaltosePeptoneCG and PeptoneCG was curious as it was expected to be similar to SabCG. It was noted that GlucoseMaltosePeptoneCG would have had 8% sugars, MaltosePeptoneCG 4% sugars, and PeptoneCG 0% sugars, and hence not likely an effect of differing osmolarity, and hence possibly some form of catabolite repression and/or possibly that maltose is less readily utilised by a sub-population of organisms.

MEA and maltose media without added glucose did have a detectable amount of glucose after autoclave sterilisation, being 7.5–10.2 mM glucose in maltose-based media, and approx. 55–100 mM in MEA, and no detectable glucose in fresh maltose in cold RO water at approximately 100 mg/mL. Hence if the MEA after autoclaving, pouring and storage had 55–100 mM glucose, and the original mass of dry malt extract powder added was 34 g/L, then 29–53% w/w of the original mass of the malt extract powder was present in the MEA as glucose. Given that the dry malt extract powder is said to typically have 60–62% w/w reducing sugars (which includes monosaccharides and the disaccharide maltose, but not sucrose nor longer chain sugars) then 48–86% of those reducing sugars were present as glucose in the MEA.

Hence there was a degree of hydrolysis of the pure maltose within the MaltoseCG and MaltosePeptoneCG media into glucose likely during autoclave sterilisation at 121°C for 15 min, plus warm-up and cool-down time, and time at approx. 50-70°C during pouring. It was originally intended that the maltose solution be filter-sterilised and added to cooled liquid agar media, but this was not the case and instead this experiment was used mainly to demonstrate that when maltose is in a media, it does hydrolyse to glucose to a physiologically significant degree during normal autoclave sterilisation given that normal human blood glucose concentration is approx. 5–10 mM. The noted glucose concentration suggested approximately 1.8 g/L maltose had hydrolysed out of 40 g/L initial maltose, or 4.5%.

## Clarified V8® Original juice, tomato juice, celery juice agar media supplements and heat-labile factor effects

There was no significant difference between media with or without various clarified vegetable juice extracts overlain over the top of SabCG media, and hence no significant heat labile factor nor missing vital vitamin or mineral or such supplied by these vegetables at least that enhances the detection of outdoor airborne fungi in mid-April (autumn), Melbourne, Australia (Figs 2 and 3; p = 0.91). It was noted that uncooked tomato juice significantly changed odour when heated, going from a 'grassy' fragrance to a 'tomato soup' odour, yet this had no discernible effect on numbers of CFU nor colony morphology, range of cultured organisms, etc.

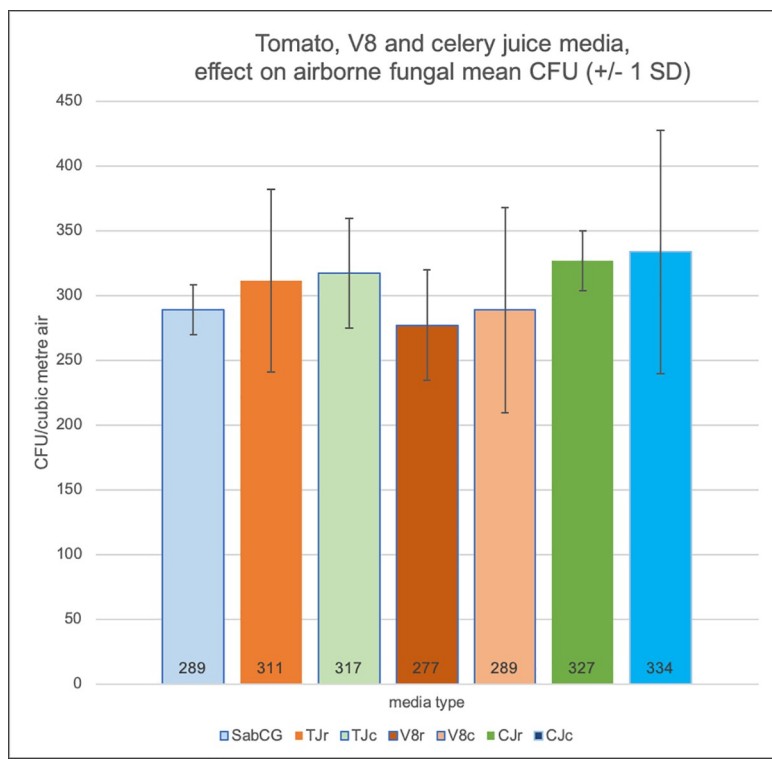

**Fig 2. Media supplemented with raw or cooked vegetable juices.** SabCG (1 mL water control); TJr / TJc = 1 mL Tomato Juice, raw / cooked; V8r / V8c = 1 mL V8 ® Original juice, raw but supplied UHT pasteurised / cooked; CJr / CJc = 1 mL Celery Juice, raw / cooked. Each medium was tested in triplicate, and all were overlain over the top of pre-prepared SabCG media in Petri dishes.

Early development cycles using vegetable juices that had not been clarified at all were found to not be useful for two reasons: being impossible to filter sterilise; being difficult to see through the tomato-based media from underneath, making enumeration and identification of

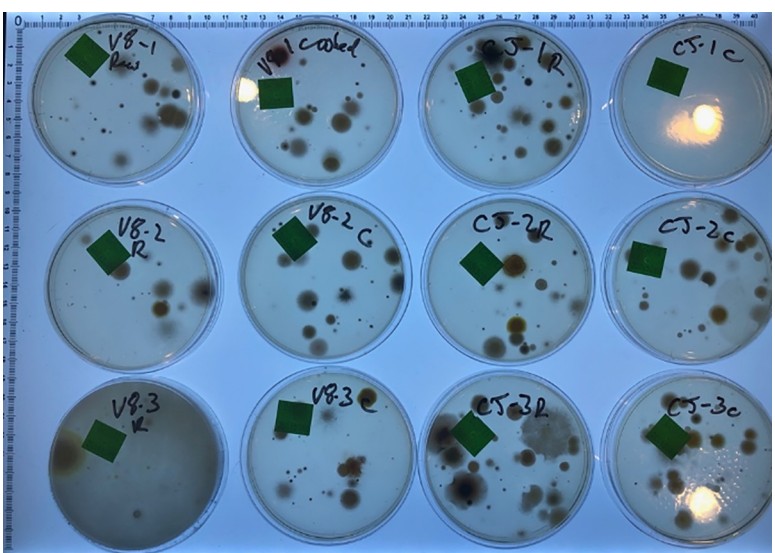

**Fig 3. Images of results of raw and cooked vegetable juices.** Images of some of the resulting Petri dishes (Fig 2) after culture for illustrative purposes only. Noted typical variation in numbers and types of fungi between each plate.

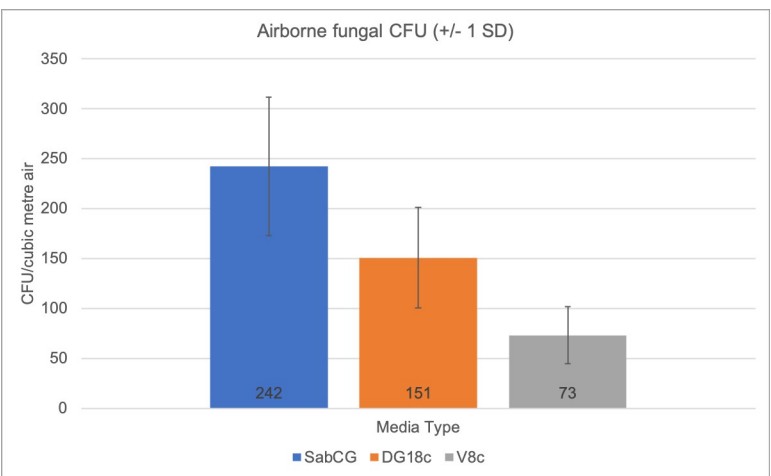

**Fig 4. Graph of V8c, DG18c and SabCG media.** 150 L outdoor air was sampled via Andersen 400-hole sampler onto six replicate plates of each of V8c, DG18c and SabCG agar media then incubated at 27˚C for 3 days.

colonies difficult. It was also determined by Lugol's Iodine solution there was a significant amount of starch in unclarified juices that could presumably affect results by selectively advantaging organisms with amylase activity, further explored in other experiments presented below.

The method of adding 1 mL of clarified juices including V8® Original juice over the top of SabCG media was useful but not equivalent to the media known as 'V8 media,' being approximately 1/3 strength and with glucose, peptone, chloramphenicol and gentamycin.

## V8c, DG18c and SabCG media

The V8c and DG18c media (containing antibiotics chloramphenicol and gentamycin, GC) were compared with SabCG via standard collection of airborne particles by 400-hole Andersen sampler outdoors in a suburban location on a winter's afternoon with a light breeze (approx. 4 km/h, 1 m/s). Six replicate plates were used per medium, and collected in 'collated' sequence, being SabCG, DG18c, V8c, then repeating in that sequence to better allow for random changes in wind speed, direction and hence likely changes in numbers and types of viable airborne fungi over the course of the experiment, being approximately 2 h (Figs 4 and 5).

There was a significant difference between the three different media regarding the number of airborne CFU/m$^3$ detected (p < 0.0003) when incubated for 3 days at 27˚C.

The mean for SabCG was significantly highest at 242 CFU/m$^3$, while DG18c and V8c were 151 and 73, respectively. Their corresponding SD were 69, 51 and 29, respectively, or 29%, 34% and 40% of their means, respectively. There was hence little overlap in their 68% confidence interval (i.e., +/- 1 SD around the mean), being 173–312, 100–201 and 45–102, respectively, assuming a normal distribution. The Kurtosis and skew of results for SabCG were 1.77 and 1.43, respectively, and for DG18c they were -1.89 and 0.66, respectively, and for V8c they were -0.54 and -0.03, respectively, and hence generally normal or only mild skew for DG18c and V8c, but significant for SabCG results, while Kurtosis was acceptable for each medium [79].

Additionally, the V8c medium was notably difficult to see through, being red in colour and nearly opaque and hence difficult to quickly observe the reverse / underside of many colonies, normally very useful in identifying/differentiating *Cladosporium* from *Aspergillus* and *Penicillium*.

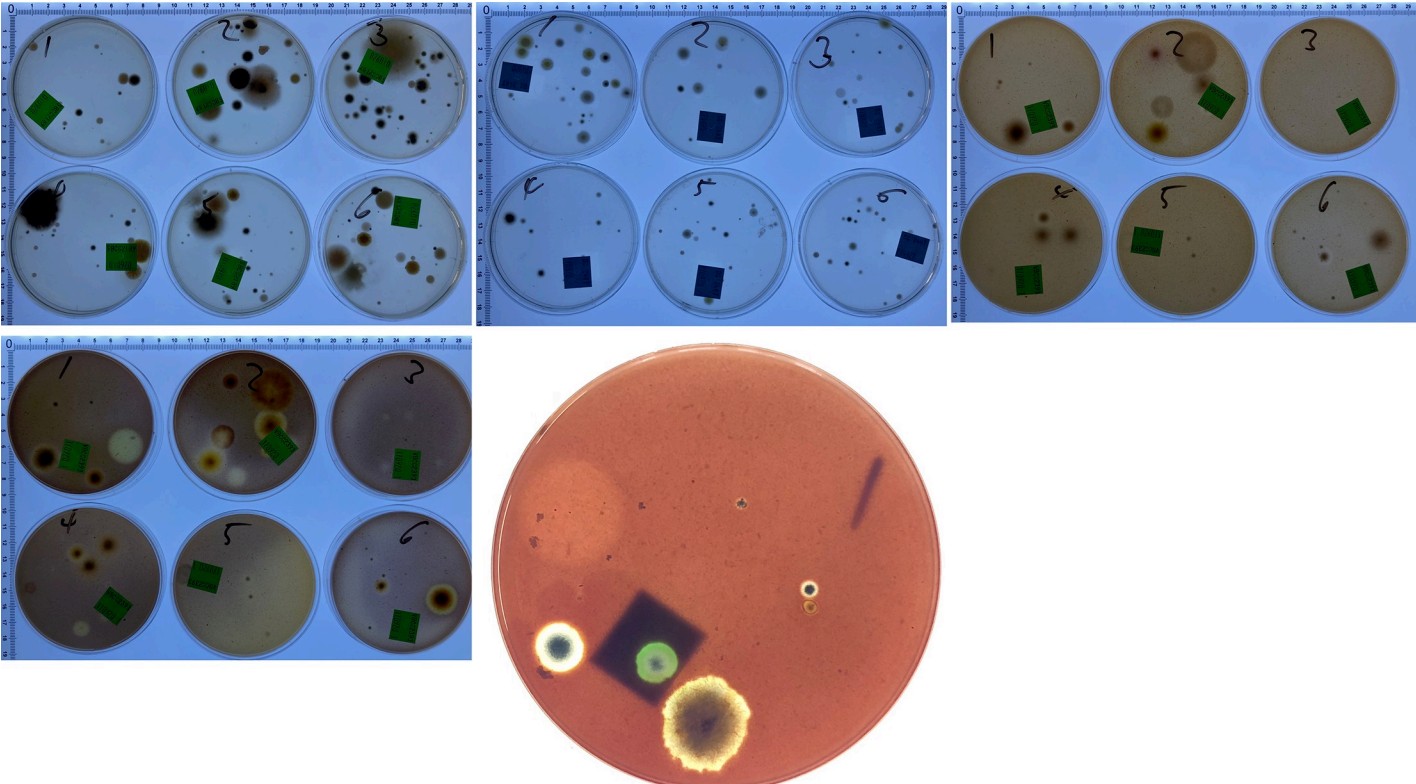

**Fig 5. Images of SabCG, DG18c and V8c media.** 150 L of outdoor air was sampled via Andersen 400-hole sampler onto six replicate plates of each of V8c (A), DG18c (B) and SabCG (C) agar media and incubated at 27˚C for 3 days; (D) V8c media stained with Lugol's Iodine, staining starch dark, seen from below the media (reverse side); (E) Detail of stained V8c media plate #1 from the upper side. Zones of clearing of starch was noted around some but not all colonies, with larger colonies being more associated with definite zones of clearing, smaller colonies without. Some seemingly colony-less zones of clearing may have been artefacts of wispy, low-mass colonies being rendered essentially invisible when the Lugol's Iodine solution was added, causing structures to lay flat against the gel surface.

It was also noted that the V8c medium had faint zones of clearing around some colonies but not all, and nearly always the colonies with clearing were large compared with colonies without zones of clearing (Fig 5E). This clearing was found to be due to the localised lack of starch in the media, as determined by flooding the plates with Lugol's Iodine that stains starch dark, and hence likely due to digestion of the starch by some but not all organisms, and the organisms digesting starch growing more rapidly than those not doing so. The standard V8 agar medium formula does not include simple sugars such as glucose, nor peptone or similar alternative energy sources in any great abundance given V8® Original juice is stated as having 3.3 g/100 mL carbohydrates, of which 2.7 g/100 mL are sugars, 0.8 g/100 mL protein and 1.0 g/100 mL 'dietary fibre.'

The DG18c medium did cause the colonies that grew to grow at a somewhat similar rate, and hence the colonies were more consistent in size at three days, but were quite often under-developed compared with SabCG, being without good maturation of spores, sporulating structures and typical colouration thus making identification more difficult and time-consuming.

It was also noted that the condensation from the DG18c was quite sticky, having significant amounts of glycerol presumably picked up while running over the gel that hence did not dry prior to, during or after sample collection. This caused one plate to become contaminated with a significant number of yeast colonies, causing it to be rejected from the data set. Typically, the condensate on the Petri dishes of other media without glycerol merely dry during the sampling

of 150 L air, as also the media itself, typically visually apparent by the 400 dimples in the gel surface corresponding to the holes in the 400-hole Andersen sampler top-plate. The pattern of dimples is useful in determining that the bottom dish with media has not rotated during sampling due to vibrations from the air-pump as this affects the statistical calculations that are based on the assumption that a hole is either negative for growth (0 CFU), or has one or more viable CFU, and hence appears positive for growth despite possibly having multiple original viable CFU deposited on the gel surface.

It is unclear if the winter season may have caused a shift towards more high-$a_w$-tolerant organisms compared to hot dry presumably lower $a_w$ seasons, and hence higher apparent numbers in the high $a_w$ SabCG medium cf. the lower $a_w$ DG18c. Melbourne, Australia tends to have fairly dry, mild winters.

The use of six replicate plates was found to be useful (compared to three) given the noted significant variation presumably due to the combination of the inherent uncertainty in the 400-hole Andersen collection method, and the uncertain nature of wind currents and weather. The use of three replicate plates was, however, not unreasonable in other experiments wherein large numbers of different media were compared simultaneously, thus generally controlling for day-to-day variation. The P-values support this conclusion.

## Media tests: MEA, PDA with/without antibiotics, and mineral supplements MS1, MS2

There was no great difference in numbers of outdoor airborne fungi collected early-September (early spring) in Melbourne, Australia, between various mineral supplements and other media such as PDA or MEA, compared with SabCG (Figs 6 and 7). The variation between media groups was generally less than the variation within each group, but having the CG antibiotics present facilitated enumeration and identification of organisms.

Chloramphenicol / Gentamycin (CG) was useful in reducing numbers of bacterial colonies to zero, increasing the confidence in the identification of yeast colonies that usually look similar (shiny, glabrous, usually small, round colonies; Fig 7). Also the fungal colonies were more regular in shape, tending to have a rounded circumference, cf. irregular colonies with 'holes' and scalloping from bacterial colonies growing where the fungal colony would otherwise be, and fungal colonies having more regular colours, appearance, sporulation/fruiting bodies, etc.,

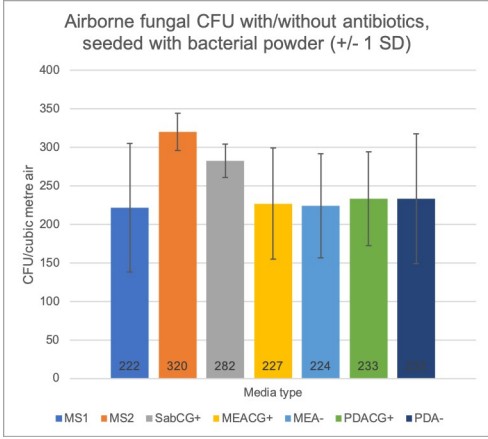

**Fig 6. Effect of antibiotics in various media vs bacteria powder challenge.** Mineral supplemented medias with CG antibiotics (MS1, MS2), other medias with antibiotics (SabCG+, MEACG+, PDACG+), and without antibiotics (MEA-, PDA-), seeded with airborne bacterial powder during sampling 150 L outdoor air, then incubated at 27˚C for 3 days.

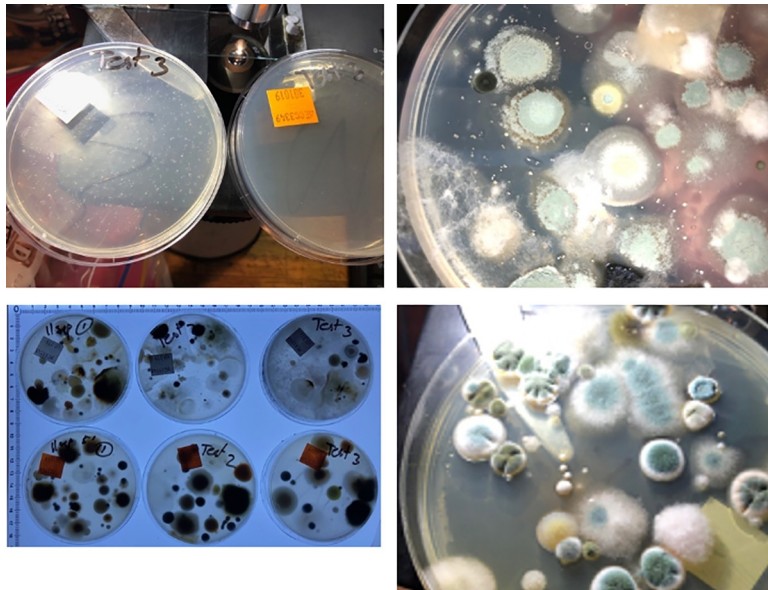

**Fig 7. Visible effects of bacterial growth on fungi.** Images of Petri dishes (Fig 6) seeded with bacteria powder while drawing 150 L outdoor air: (A) at 24 hr, 27˚C, MEA (left) shows many bacterial colonies in the pattern of the 400-hole top-plate, and MEACG (right) showing no colonies due to CG antibiotics. Fungal colonies are not typically visible until 48 hrs, and not typically enumerated nor identified until 72 hr; (B) Example of MEA after 72 hr, 27˚C, showing the small bacterial colonies in the 400-hole plate pattern. Fungal colonies were typically somewhat different to colonies on MEACG, having visibly different morphologies and colours, e.g., *Penicillium* spp. colonies were typically flatter, smaller, with scalloped edges and the spores were pale and at a delayed state of growth/development, while other fungi remained very sparse, spreading without apparently sporulating and hence making identification very difficult; (C) Example of results at 72 hr, 27˚C, with MEA (top row) and MEACG (bottom row), with notably different colony colours and morphologies with/without CG antibiotics; (D) Example of typical colony morphologies on MEACG and other media with antibiotics such as SabCG, having better maturation, conidia development, colouration and overall more consistent colony shape thus aiding identification and enumeration.

presumably due to not having to actively respond to competing bacteria nearby, or passively via the drain on available resources in the local media.

Mineral supplements MS1 and MS2 caused the media to go cloudy, which was less than ideal for counting and identification purposes. This cloudiness occasionally lessened over time and/or occasionally when colonies grew nearby, forming halos of clear areas presumably due to changes in pH due to atmospheric $CO_2$ and/or biological processes and fermentation products also including $CO_2$, and also likely ammonia, organic acids, etc.

## Various glucose concentrations of media vs airborne fungal detection

There is some small degree of difference in numbers of CFU (and their colony morphology) between SabCG media with different concentrations of glucose, with possibly a more ideal concentration being about 2% cf. the standard 4% (Figs 8 and 9). The effect is small, however, and would be unlikely to significantly influence counting or identification. There is little effect over the tested range from 1% to 8%, as predicted from the estimated $a_w$. There was some effect noted in the PeptoneCG medium, but even then it seems the majority of airborne fungi able to grow at 27˚C within 3 days are able to substantially grow and sporulate without a sugar source, using the peptone as an energy and nitrogen source to complete their life-cycle, at least when sampled in early-February (summer), Melbourne, Australia. Glucose seemed to generally be of benefit, but some experiments suggested a possible suppression of growth at high sugar concentrations, initially hypothesised to be a catabolite-repression and/or high osmolarity effect.

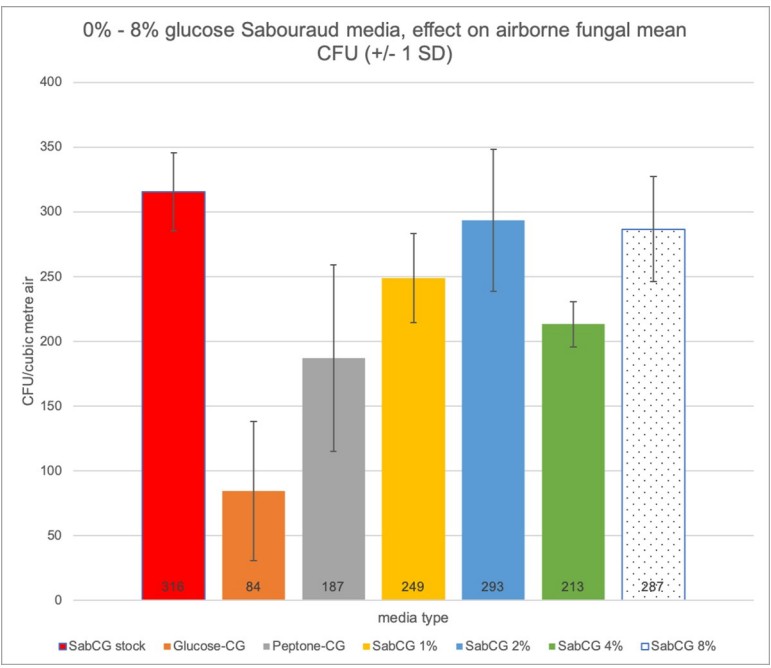

**Fig 8. Various media glucose concentrations graph.** Graph of CFU/m$^3$ air (from 150 L air sampled) vs media with various glucose concentrations. SabCG stock is 4% glucose. The colonies on the GlucoseCG (no peptone, 4% glucose) medium were very under-developed and not strictly comparable with those on other media. Colonies on PeptoneCG were similar to those on SabCG 1%, 2%, 4% and 8% glucose.

## Outdoor CFU/m$^3$ by season detected by SabCG

Each experiment performed included a control, being stock SabCG and hence it was possible to collate the data from multiple experiments conducted on different days, which happened to be different seasons (Table 1).

Given the experiment presented in Fig 4 (winter) had six replicate plates rather than the three replicate plates used in the four other included experiments (spring, summer, autumn) the use of the mean of the means prevented over-representation of the winter data in the over-all results that would otherwise be 274 CFU/m$^3$. Thus the weighted mean of the pooled standard deviations (Sp) is 50 CFU/m$^3$.

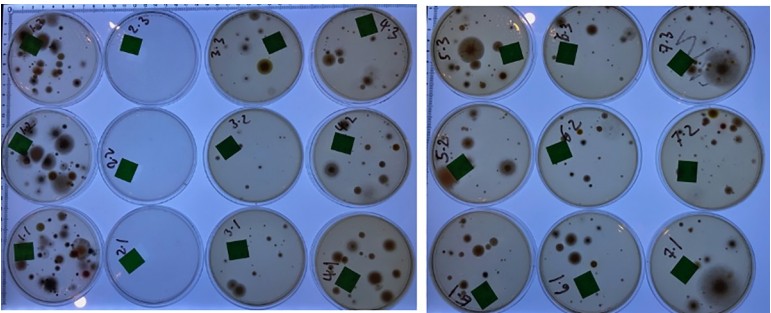

**Fig 9. Images results of various glucose concentrations.** Images of Petri dishes after culture as used in Fig 8: (A) Column at leftmost, Stock SabCG; second left, GlucoseCG (colonies virtually invisible and very under-developed), third-left, PeptoneCG, fourth-left / rightmost, SabCG 1% glucose; (B) Column at leftmost, SabCG 2% glucose, second-left, SabCG 4% glucose, third-left / rightmost, SabCG 8% glucose.

This excluded experiments performed in December 2019 and January 2020 (both being in summer), which were abnormally still and windy days, respectively, and an experiment performed using only 60 L outdoor air that showed too few CFU per Petri dish to be statistically useful and hence repeated using 150 L.

### Flame sanitisation of 400-hole Andersen air sampler top-plate

Dousing the aluminium Andersen 400-hole air sampling top-plate with alcohol and flaming it did effectively sterilise or at least sanitise it of significant numbers of dry viable *Penicillium* spores placed there (Fig 10). This was not entirely expected because it was presumed the heat would be insufficient to raise the temperature of the metal above that required for significant killing for long enough to do so. It is known that aluminium has a high thermal conductivity and is often used in heat-sinks and cookware. It was also noted that some charred debris and possibly inorganic grit was often left behind, however, which seemed to accumulate in the quite narrow holes, reducing air flow and the number of 'open holes,' critical to the operation of the Andersen 400-hole sampler.

## Discussion

The use of SabCG as commonly formulated and commercially available was found to be reasonably consistent and sensitive for the detection, enumeration and identification of airborne

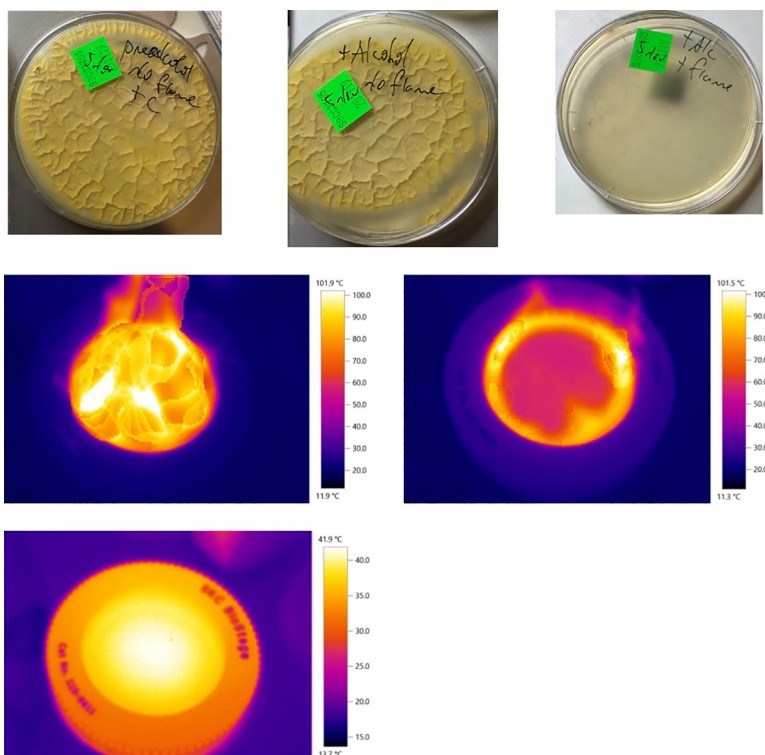

**Fig 10. Flame-sanitisation of Andersen 400-hole top-plate.** (A) Photograph of results of samples taken by swab from SKC BioStage 400-hole Andersen top-plate doped with an excess of viable dry *Penicillium chrysogenum* spores; (B) Sample after top-plate was liberally doused with common household methylated spirits but not ignited; (C) Sample after top-plate was doused with alcohol and then ignited then allowed to self-extinguish and cool for 1 minute before sampling; (D) Thermographs of flaming SKC BioStage 400-hole Andersen top-plate: top-most image was at the time of ignition (maximum estimated temperature 101.9°C); image second from top was 13 s after ignition and continuing to burn; image third from top / at bottom as 35 s after ignition at approximate time when the flames had ceased. It was noted that the temperature of the aluminium top-plate was less than 42°C at this time and was only slightly warm to the touch when repeated on other occasions without added mould spores.

viable fungi including a variety of moulds and yeasts found outdoors, at least in suburban Melbourne, Australia. This was used as a baseline / proxy for the indoor and general environment, and in the context of a relatively inexpensive, reasonably rapid and difficult to mis-read test. The medium is non-toxic and simple, having an amino-acid/peptide source widely used in microbiology and a glucose source, and supports the growth of a wide range of organisms found in outdoor air, which do not require growth factors or such from sources such as malt or vegetable juices, and is not selective for amylase- or maltase-positive organisms, and a wide range of $a_w$ requirements, which is useful for its intended purpose in estimating how mouldy a house is, especially those that are damp.

The inclusion of anti-bacterial antibiotics chloramphenicol and gentamycin appears to improve fungal colony morphology, colouration and rate of development by suppressing bacterial growth and hence making enumeration and identification faster and easier even for more experienced staff, including simple observation to determine if it is a yeast or bacterium given their colonies often look very similar.

Detection and enumeration of early-coloniser fungi such as *Penicillium*, *Aspergillus*, *Alternaria*, *Ulocladium*, *Rhizopus*, *Mucor* and yeasts may be a useful proxy for the general degree of mouldiness of a house as indicator organisms, being always present in low numbers outdoors (thus a useful control for the sampling equipment, media and culture conditions), growing rapidly and easily, are relatively easily enumerated and identified, are present in significantly elevated numbers in water damaged buildings, and may indeed cause respiratory disease directly.

Further experiments are planned to better analyse statistical aspects of the original 400-hole air sampling method published by Andersen in 1958 [18], as well as more meaningful analyses of real-world houses with and without known mould/moisture issues, and eventually ideally finding any hypothetical correlation between mould and reported symptoms by occupants. This has been especially difficult with the great variety of different media and methods used historically, coupled with a lack of clarity surrounding the different objectives and practical considerations of estimating the mouldiness of an inhabited building, compared with more well-known but significantly different concerns for testing foodstuffs, food-preparation surfaces, surgical and manufacturing clean-rooms, pathology samples including medical, veterinary, plants, etc.

The use of malt-extract-based and/or starch-bearing agar media, often used for the detection of plant pathogens and the contamination of plant-derived foodstuffs, is of questionable suitability for detecting organisms saprophytically degrading organic materials found in damp houses such as carpet, paper, cardboard, plasterboard, timbers, etc., or natural micro-environments such as leaf litter, fallen logs, grass, or animal materials such as hair, wool, fur, skin flakes, leather, dander, etc. This is possibly because such household dusts and materials are unlikely to have significant amounts of starch or maltose compared with foodstuffs and germinated and/or rotting grains.

Maltose does hydrolyse over time and temperature, reportedly approximately 5% or so at 120˚C for 1 hour [80], and supported by the results indicating approx. 4.5% after an autoclave cycle of 1 L liquid media. Malt and hence malt-extract is highly variable, being a pivotal aspect of brewing beer and whisky (or whiskey) typically using barley but possibly wheat or rye and germinating them under differing conditions to cause the starch to enzymatically break down into various sugars including maltose, and then may be roasted and even smoked to impart a variety of flavours before further processing and extraction typically including concentration by boiling and evaporation, all of which have different, variable and/or un-reported durations and conditions of heat treatment. This would likely lead to significant regional and batch-to-batch variation that is not well controlled or described and may or may not have a significant

effect on sampling results when attempting to compare them between groups using different media suppliers, autoclave conditions or working in different countries and/or times.

It seems that a significant proportion, and possibly the majority, of reducing sugars present in the original malt extract powder added to MEA is then present as glucose after autoclaving and storage, although it is unclear how much of the glucose was present in the powder itself from the production of malt extract, primarily by prolonged boiling at a somewhat acid pH and likely leading to maltose hydrolysis.

Regardless, it appeared that the presence of maltose when added as pure powder to MaltosePeptoneCG and MaltoseGlucosePeptoneCG agar media suppressed the numbers of CFU/m$^3$ detected in outdoor air compared with GlucosePeptoneCG (= SabCG), although it was unclear exactly why.

Estimates of the nitrogen content of MEA medium using 34 g/L malt extract (Oxoid LP0039) are approximately 0.0374% w/v total-N, 0.0204% w/v amino-N, and hence approximately 39% the total-N and 70% the amino-N content of Sabouraud media prepared with 10 g/L Mycological Peptone™ (Oxoid LP0040). Given that the results suggest the presence of peptone (and hence nitrogen) is a more critical factor than the presence of glucose or maltose, it seems reasonable to add it in a specific manner in relative abundance as per the Sabouraud medium rather than the lesser amount in the MEA medium.

Logically and practically it seems better to reduce batch-to-batch variability by simplifying the media using chemically-pure glucose, and a consistent nitrogen source such as Mycological Peptone™, and using reliable stable antibiotics such as chloramphenicol and gentamycin, as per SabCG. This also avoids the noted lower detection of outdoor airborne CFU/m$^3$ when maltose and/or malt extract was used in media such as MEA, MEACG, MaltosePeptoneCG, etc.

The notion that the high $a_w$ of the SabCG media might suppress the apparent numbers of outdoor airborne fungi by preventing the growth of numerous xerophilic organisms present or other possible causes is not supported by the data when the low $a_w$ media, DG18 (with antibiotics) was compared with SabCG.

Other workers had noted a reduction in the viability of some common fungal spores grown under low $a_w$ then exposed to high $a_w$ media, putatively due to an osmotic-shock effect causing the spores to swell and lyse [38,41], thus reducing the apparent numbers of airborne fungi recovered on high $a_w$ media. This is curious given that few common fungi are markedly inhibited by high $a_w$ [81], and because the likely highest contributors to fungal growth are high $a_w$ materials/environments, which is the notable problem in a damp building. This is presumably quite different to the problems of low $a_w$ food spoilage by xerophilic/xerotolerant organisms. Of course, in assessing a building for mould in practice it is to best achieve a reasonable compromise between detecting the full range of viable organisms present, or the subset of 'indicator organisms' virtually guaranteed to be present if the building is or has recently been damp and thus mouldy, and to do so reasonably consistently, rapidly in culture, and facilitating enumeration and identification.

SabCG appeared to achieve a reasonable balance of this under the experimental conditions used, and using outdoor airborne fungi as a proxy for the range of organisms found in buildings.

The V8c agar medium (with antibiotics) significantly yielded the lowest numbers of outdoor airborne fungi when compared with SabCG and DG18c media in winter in Melbourne, Australia. This was interesting but not unexpected given its relative lack of simple sugars and amino acids. That the largest colonies found on the V8c agar medium were always associated with a zone of clearing of the starch granules under and around it was interesting, especially when very small colonies were generally not associated with a zone of clearing. This suggests

that the large colonies are able to grow because they were digesting the starch and hence at a considerable advantage in the otherwise relatively energy-poor medium.

The generally-cited incubation conditions for DG18 (25°C, 5–7 days) presented some challenges given that this is often more than the time required for many common moulds to grow to maturity, sporulate and have progeny colonies of a size and state of maturity making them appear to be the originally collected generation, albeit usually smaller but tending to appear like other slower-growing organisms, thus adding a source of bias and confusion.

Similarly, other sparsely-growing or wispy organisms tended to spread avidly and hence cover smaller colonies, obscuring them and making enumeration and identification difficult.

The longer time of incubation also presents a problem when there is a potential health-risk at a likely mouldy house, office, etc., when the time for results turnaround is important. Hence the use of 3 days incubation at 27°C as standard appeared to be a reasonable compromise, being warm enough to allow the reasonably rapid growth of many organisms, but not so warm as to inhibit temperature-sensitive organisms such as *Penicillium* and *Cladosporium* species commonly found in damp houses. Many environmental organisms including some strains of plant pathogens *Eutypa lata* and *Botryosphaeria* do not grow well in culture at temperatures much above 20–24°C depending on the climate they were isolated from [82].

In testing various media, it was tempting to use more controlled conditions such as filtered air intentionally seeded with known species of cultivated moulds, but it was thought it would be a better test of the natural world to use the likely wider range of wild-type organisms found outdoors. Additionally, in testing houses and other buildings for mould, the outdoor air is always tested and compared as a control given that non-mouldy / non-WDB typically have a similar number of airborne as outdoors, but significantly mouldy / WDB have more than outdoors, albeit typically of a narrow range of organisms that grow rapidly in damp conditions on building materials. Therefore, the more important concern is to reliably and rapidly detect the likely relative shift in the range of organisms differentially rather than exhaustively and absolutely.

When the results of the number of outdoor airborne viable fungal CFU/m$^3$ detected by culture on standard SabCG medium were compared between each different season (excepting days of very low or high wind speeds) it was found that the mean of the mean results was 282 CFU/m$^3$ (Table 1) with a weighted mean of the pooled SD (Sp) of 50 CFU/m$^3$. Hence the 68% confidence interval (C.I.) was 232–332 CFU/m$^3$, and the 96% C.I. was 183–382 CFU/m$^3$. This yields a coefficient of variation of 18%.

It is interesting to note the calculated C.I. as these approximate values are often seen in practice when sampling outdoor air as a control prior to entering a building under assessment for mould, excepting adverse or unusual weather events including rain, strong winds, or the air sampling unit being positioned too close to or downwind of a notably mouldy building or materials removed from one, or some types of trees and wetland areas via personal observation of many hundreds of sampling occasions over many years and locations (results not shown). Further routine studies with greater numbers of replicates are planned, and during various weather conditions and seasons using only SabCG to simplify such studies down to a more manageable degree. Sampling notably mouldy houses using DG18c and SabCG suggested the high $a_w$-requiring *Trichoderma reesei* was occasionally present but unable to grow on DG18c (pilot studies, results not shown).

While it is possible to flame-sterilise or sanitise the 400-hole Andersen impactor top-plate, it is of questionable advantage to do so while onsite, likely having several orders of magnitude less than 1% an effect on the results even if taken from a very mouldy location to a sterile one, especially compared with the typically far greater natural sampling uncertainty.

Flaming the top-plate onsite carries some practical considerations such as transporting and carrying flammable liquids and setting fire to it between uses, frequently while wearing flammable gloves and disposable polyethylene overalls while carrying flammable plastic bags, and often in environments with large amounts of sawdust, cardboard particles, construction materials and waste, plastic sheets used for containment cells during remediation works, paints and thinners, and sometimes quite strong air currents either outdoors, or from outdoors in damaged buildings, or from blowers, air-movers, fans, air-filtering units, heaters, air-conditioning units, dehumidifiers, etc., within buildings often without functional smoke alarms, fire-fighting equipment and functional fire suppression sprinkler systems, proper fire escapes, or often floors, stairs and power.

Previous experience found that collection efficiency was impaired when the top-plate was wiped with cleaning solutions that left any residue such as benzalkonium chloride and detergents, and hence the regular use of hot RO water and an ultrasonic bath was implemented to clean the holes, keeping them open and keeping air flow consistent between uses as per the manufacturer's instructions [19].

Similarly, it was also found that the number of CFU detected by the 400-hole sampler was not significantly affected by having been used previously in a very mouldy location, such as sampling a 'clean' location immediately after a location with high numbers of viable airborne moulds (personal observation).

This was not surprising given that the holes are 0.25 mm diameter, and approximately 1.5 mm deep. Thus all 400 holes together have a collective void-volume of less than 30 μL. Hence, for there to be a reasonable chance of increasing a subsequent air sampling run by one single CFU there would have to be one CFU within the 30 μL of void-volume from the previous air sampling run, which would therefore be $3.3 \times 10^7$ CFU/m$^3$, which is very far beyond the typical 282 CFU/m$^3$ found outdoors normally, and even far beyond the lower limit of the highest risk category commonly cited (5,000 CFU/m$^3$) by many orders of magnitude.

Prudence and habit, however, meant that the 400-hole top-plate was wiped top and bottom with a commercially available single-use disposable lens cleaning wipe that comes pre-soaked with isopropanol to remove dusts rather than sterilise or sanitise the top-plate. This is reflected in sampling protocols for various airborne microorganisms including fungi and bacteria via the same Andersen sampling apparatus, using isopropanol merely to clean the sampling plate without setting fire to it according to the manufacturer's instructions [19,83], and ideally the pump unit, hands/gloves and other equipment potentially exposed to mould and dust.

## Supporting information

**S1 File. Tables of raw data.** These tables include CFU/plate counted, calculated CFU/m$^3$ air via Andersen, 1958 with a 1.25x adjustment to the raw CFU/plate given polymer Petri dishes were used, and general categories of genera as adapted from ASTM D7391-20 section 12.3.2 [78], pertaining to Figs 1–9 and Table 1.
(DOCX)

**S2 File. Tables, data analysis and graphs.** A MS-Excel notebook of several spreadsheets pertaining to Figs 1–9 and Table 1, including ANOVA analysis, means, standard deviations, and charts/graphs, and also an adaptation of Andersen 1958 Table 1 adjusted to convert CFU/plate to CFU/m$^3$ with the 1.25x adjustment factor for use of plastic Petri dishes.
(XLSX)

## Acknowledgments

Many thanks are given to all the staff at the Media Preparation Unit, The Peter Doherty Institute for Infection and Immunity, The University of Melbourne, Parkville/Melbourne, Victoria, Australia, with particular acknowledgments (in no particular order) to Elena Paraskeva, Kim Lai Bell, Claire Fraser and Elizabeth Trajcevska, with humble apologies to anyone I have missed.

## Author Contributions

**Conceptualization:** Wesley D. Black.

**Data curation:** Wesley D. Black.

**Formal analysis:** Wesley D. Black.

**Investigation:** Wesley D. Black.

**Methodology:** Wesley D. Black.

**Project administration:** Wesley D. Black.

**Resources:** Wesley D. Black.

**Software:** Wesley D. Black.

**Supervision:** Wesley D. Black.

**Validation:** Wesley D. Black.

**Visualization:** Wesley D. Black.

**Writing – original draft:** Wesley D. Black.

**Writing – review & editing:** Wesley D. Black.

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
