## [Decision Letter · Decision Letter 0]

30 Sep 2020

PONE-D-20-26205

A comparison of several media types and basic techniques used to assess outdoor airborne fungi in Melbourne, Australia.

PLOS ONE

Dear Dr. Black,

Thank you for submitting your manuscript to PLOS ONE. After careful consideration, we feel that it has merit but does not fully meet PLOS ONE’s publication criteria as it currently stands. Therefore, we invite you to submit a revised version of the manuscript that addresses the points raised during the review process.

We look forward to receiving your revised manuscript.

Kind regards,

Zonghua Wang, Ph.D.

Academic Editor

PLOS ONE

Journal Requirements:

"The author has declared that no competing interests exist."

We note that one or more of the authors are employed by a commercial company: Biotopia Environmental Assessment.

2.1. Please provide an amended Funding Statement declaring this commercial affiliation, as well as a statement regarding the Role of Funders in your study. If the funding organization did not play a role in the study design, data collection and analysis, decision to publish, or preparation of the manuscript and only provided financial support in the form of authors' salaries and/or research materials, please review your statements relating to the author contributions, and ensure you have specifically and accurately indicated the role(s) that these authors had in your study. You can update author roles in the Author Contributions section of the online submission form.

2.2. Please also provide an updated Competing Interests Statement declaring this commercial affiliation along with any other relevant declarations relating to employment, consultancy, patents, products in development, or marketed products, etc.  

Reviewers' comments:

Reviewer's Responses to Questions

**Comments to the Author**

1. Is the manuscript technically sound, and do the data support the conclusions?

Reviewer #1: Yes

Reviewer #2: Partly

2. Has the statistical analysis been performed appropriately and rigorously? 

Reviewer #1: I Don't Know

Reviewer #2: No

3. Have the authors made all data underlying the findings in their manuscript fully available?

Reviewer #1: Yes

Reviewer #2: Yes

4. Is the manuscript presented in an intelligible fashion and written in standard English?

Reviewer #1: Yes

Reviewer #2: Yes

5. Review Comments to the Author

Reviewer #1: In this manuscript by Black used several media for the detection of airborne fungi including malt-extract agar (MEA), Sabouraud dextrose agar (Sab, SDA, SabCG), potato dextrose agar (PDA) with and without antibiotics chloramphenicol & gentamycin (CG) and compared for their suitability in detecting a range of common airborne fungi. The author concluded that there is a little variation in mean numbers of colony forming units (CFU) and types of fungi recovered between all media except between SabCG, Dichloran-18% glycerol (DG18) and V8® Original juice agar media which showed a significant difference. Also concluded that SabCG can be used as a standard medium. The manuscript is well written.

For the most part, this manuscript can provide relevant results in the fungal biology field. However, this manuscript should need to be revised before it is published.

- Although the journal PLOS ONE does not have a strict limit about paper length, your paper needs to be shortened. The paragraph is too long, some data are redundant, some sentences are too long and confusing. I suggest shortening and reviewing the full paragraph.

- The figures are not clear. Some pictures edges are missing, I suggest you present your pictures in appropriate way to make it clearer for readers.

Reviewer #2: - First, I would like to thank the author for the manuscript entitled “A comparison of several media types and basic techniques used to assess outdoor Airborne fungi in Melbourne, Australia”

- The author tries to compare between widely used media in the detection of airborne fungi, the paper suggests SabCG media as the most detector media for such kind of fungi. The author compares between theses known media and conduct his experiments with different stages and seasons with his own goal for each experiment. He tries to find if any additives or enrichments may increase the ability of detection, identification, and enumeration of airborne fungi, and also he compare between media which can detect xerophilic fungi at low water activity and SabCG media, also he test the CFU detection, identification, and enumeration in case of bacterial contamination, in addition to other experiments.

- Overall I think the research paper needs more organization, add / update some references, clarification of some statistical analysis results, and the following are my comments on it:

In the introduction part,

- Lines 33-36 please add a reference for the information related to acceptable limits of airborne fungi (normal/ maximum). You said there's a lack of widely-accepted limits for maximum permissible and/or normal exposure to occupants, or even what may constitute a ‘moldy’ house. And I think you can find such information’s, Please see this review https://doi.org/10.1080/10473289.1996.10467526 may it will help.

In the same paragraph line 34, I have a question, please? What is the difference between identifying fungi within buildings and other places? If the difference exists please mention it, I think you are right with the point of no widely accepted methods for detection and enumeration since no one exactly knows the best method and if you change the method you will get different results. But at least for fungi-identifying, i think we all following the same protocols.

Finally in this paragraph, and about the relation between damped houses and molds, the Ref. 32 in your paper defines or set the correlation between molds and damped houses as you mentioned later.

- Line 57-58, fungal growth on building materials always occurs as a result of a high water activity (aw) at the surface of the material for most types of fungi, The minimal water activity for supporting fungal growth was examined by several authors indicating aw-min. values around 0.67–0.75, but growth also depends on temperature, time and the material .and fungus also needs water for 2nd metabolism and the production of mycotoxins which depends on the substrate, aw, temperature, and other environmental factors. Please see Nielsen KF et al. (1999). Production of mycotoxins on artificially and naturally infested building materials. Mycopathologia, 145:43–56.

- Line 173, (c.f.) refers to what? Please write the word/s from which the abbreviation was made when write it for the first time in the text.

- In materials and methods part,

- General comments:

- I think it’s better to add more details for all media you have been used, plus the mixed ones with/out supplements or enrichment/s, all should be included in details, e.g., (glucose-Maltose-peptone media and maltose peptone (please more clarifying in the text ). Details of adding supplements and media enrichments should be presented more clearly in this section instead of the results section later.

Line 303, (2-5 knots) please, use SI units or put the equivalent SI unit wit it.

- Add the method you used to identify the detected fungi to your methods.

- Results part:

- Line 336, please, where is malt extract media in your results and also in figure (1)? You tested here the maltose utilization in three Maltose based media and also peptone based media. I think the first sentence should be changed to be “Testing various peptone and maltose-based media (Fig 1) in late-January 2020 (summer) in Melbourne, Australia” If I am not mistaken, your results not only showing the maltose utilization but also peptone utilization.

- Please add all the formulas of the media you have test in Materials and methods.

- Line 356, figure (1).I think it’s good to show your statistical analysis data for variation between and within groups in one table in text, not only in supplementary tables.

- In supplementary Table S1.1 (Fig 1): the sample is the total numbers of Identified fungi by collecting a 150 liters sample of air, and CFU/m3 calculated by the equation 1.25( samples detected from 150 L/0.15) but not all CFU/m3 numbers were calculated well, for example, the replicate 2 in media 3 detect 54 fungi with 150L so the CFU/m3 should be 450 but you complete your analysis with 500 CFU/m3 ??? Could you please clarify this? And also why is the big variation in detection within the group, What you think?

Line 358, the author said (The results suggested that maltose was not commonly utilized by the fungi sampled from the outdoor air, being similar to Peptone CG despite a noted degree of hydrolysis of maltose to glucose) could you please show similarity relation between both?

- Line 361, I think both carbohydrates (C) source” and (N) both are very important for fungal growth, good sources like peptone will cause excessive growth with knowing that, the C:N ratio is dependent to each fungus type. DOI: 10.1016/j.mycres.2006.07.019.

- Line 387, no statistical information

- In supplementary Table S1.2 (Figs 2 and 3): please add the significance level and statistical analysis.

- Line 396, Fig 3, the reflection of light on the image affects the quality and 1st Replicate of celery juice (cooked) ??? very low detection rate with no any comparison with other replicates???

- Line 406-408: it’s better to talk about this later, after the comparison between V8c, DG18c and SabCG media or in discussion, with hence that the conditions were not the same.

- Line 412, please use SI units.

- Line 418-420 Fig4, please show your statistical analysis in one table.

- Fig 5, (a) vc8, (b) DG18,(c) Sab. By checking the images, I noticed that the number of colonies in vc8 media which was mentioned as a poorly detector is higher among others?

- Line 542, Flame sterilization/sanitization of 400-hole Andersen air Sampling top-plate, please add the details in material and methods.

- Finally, I think more organizing of the manuscript, doing the comparison between Sab, DG18, V8, MEA, PDA, etc., in the same conditions and different seasons with more replicates , because variations within each media are high, and showing the new results data with other experiments that author already conduct will be useful.

6. PLOS authors have the option to publish the peer review history of their article (what does this mean?). If published, this will include your full peer review and any attached files.

Reviewer #1: No

Reviewer #2: No

---

## [Author Response · Author response to Decision Letter 0]

22 Oct 2020

Please see the attached Rebuttal Letter / Response to Reviewers document.

I have included a copy below:

Rebuttal Letter / Response to Reviewers – PONE-D-20-26205, EMID:342782706afd3899

WD Black

22 October 2020

Rebuttal Letter / Response to Reviewers – PONE-D-20-26205, EMID:342782706afd3899

WD Black

21 October 2020

In response to email 1 Oct 2020 from Dr Zonghua Wang PhD, Academic Editor, PLOS ONE. 

Responses interspersed.

Journal Requirements:

"The author has declared that no competing interests exist."

We note that one or more of the authors are employed by a commercial company: Biotopia Environmental Assessment.

2.1. Please provide an amended Funding Statement declaring this commercial affiliation, as well as a statement regarding the Role of Funders in your study. If the funding organization did not play a role in the study design, data collection and analysis, decision to publish, or preparation of the manuscript and only provided financial support in the form of authors' salaries and/or research materials, please review your statements relating to the author contributions, and ensure you have specifically and accurately indicated the role(s) that these authors had in your study. You can update author roles in the Author Contributions section of the online submission form.

Amended Funding Statement (WDB):

There was no external funding body that provided support for this study. Instead, the sole author, WDB, self-funded their own salary and materials purchased through their own business/company, Biotopia Environmental Assessment Pty Ltd., which does not employ any other persons and has no vested interest in the outcomes of the study, its design, data collection, analysis, decision to publish nor preparation of the manuscript, aside from wishing to review, test, validate, formalise and communicate common techniques used by workers in this field, and hence which techniques the author and hence the private business/company ought to use in assessing mainly buildings for mould, and to better estimate their limitations such as uncertainty. The specific role of the sole author are articulated in the ‘author contributions’ section.

In response to the email from the Editor, Kirstin Darroch, ~21 October 2020, suggesting a further amended Funding Statement, I have included the following in the relevant section of the manuscript:

“Biotopia Environmental Assessment Pty Ltd provided support in the form of salary and materials to author WDB. The specific roles of this author is articulated in the ‘author contributions’ section. The funder had no commercial or vested interests in study design, data collection and analysis, decision to publish, or preparation of the manuscript. No additional external funding was received for this study.”

I have slightly altered the text because I am both the author and the owner of Biotopia, the funder, so it seems illogical to say that the funder had no role in the study design. Also, I purchased my own research materials through Biotopia. 

I absolutely and honestly declare that the revised Funding statement as above and now in the manuscript is true and accurate. 

2.2. Please also provide an updated Competing Interests Statement declaring this commercial affiliation along with any other relevant declarations relating to employment, consultancy, patents, products in development, or marketed products, etc. 

WDB: Funding statement / competing interests statement:

I am a little uncertain what is required here, hence an explanation is given below for your kind review: 

I am the CEO and owner of Biotopia Environmental Assessment Pty Ltd. There are no other employees/directors, staff, etc. The company was originally registered as a sole-trader in September 2011 until its incorporation in January 2019 for tax and legal-protection reasons. It’s really just me working as an independent consultant for various entities including homeowners, remediators/cleaners, builders, insurance companies, law firms and their clients, etc., in a piecemeal/contractor fashion. That is, with separate contracts rather than an ongoing nature with any client/employer. This suits us all as it allows me to be without conflicts of and/or vested interests, and this is important for legal cases especially. 

There is no funding for R&D as such other than my own salary within the sole-operator/owner consultancy business, and the costs of the research projects such as reagents, media, etc., are claimed as business expenses. The company is too small to qualify for Australian Government R&D incentives presently, and without a number of publications in this field independent grants are unlikely and hence were not sought. Hopefully this will change with new government R&D incentives, this publication and another I am in the process of writing up and intend submitting to PLOS ONE regarding the validation of a novel fungal spore staining method for total-counts of airborne and surface samples. 

I had originally performed a series of in-house verification experiments early in 2012 to determine which media was best for common indoor / outdoor airborne moulds, which the submitted manuscript’s experiments are largely based on but elaborate further. 

Additionally a number of questions were raised generally by fellow workers in this field regarding various other media and incubation conditions, and the manuscript also attempts to address them. Ideally the results presented in this manuscript will form the basis of further, more specific experiments to further determine uncertainty and correlation between true numbers of particles and the numbers detected by these methods. 

The only role this commercial affiliation played was to find and verify a reasonable media to quickly and reliably estimate the number and general types of common viable moulds/yeasts in a building or similar built structure, as distinct from food-spoilage organisms or animal/plant pathogen detection, etc. 

No particular product is endorsed or promoted, nor their manufacturer or supplier. The findings of the study are presented merely to edify fellow workers in the same field and to work towards more meaningful tools and standards in this area. 

As for the amended Funding Statement, the newly amended and inserted Conflicts of Interest statement is now as below:

“The author has read the journal's policy and has the following competing interest: WDB is the CEO and owner of Biotopia Environmental Assessment Pty Ltd (http://www.biotopia.com.au/). There are no patents, products in development or marketed products associated with this research to declare. This does not alter the author’s adherence to PLOS ONE policies on sharing data and materials.”

I have slightly altered the suggested text as per the email from the Editor, Kirstin Darroch, ~21 October 2020 to include ‘www’ in the URL, and changed the plural “we” to the singular “the author’s” as there is only one author, me. 

I absolutely and honestly declare that the revised Conflict of Interest statement as above and now in the manuscript is true and accurate. 

~~

[section deleted for brevity]

 

5. Review Comments to the Author

Reviewer #1: In this manuscript by Black used several media for the detection of airborne fungi including malt-extract agar (MEA), Sabouraud dextrose agar (Sab, SDA, SabCG), potato dextrose agar (PDA) with and without antibiotics chloramphenicol & gentamycin (CG) and compared for their suitability in detecting a range of common airborne fungi. The author concluded that there is a little variation in mean numbers of colony forming units (CFU) and types of fungi recovered between all media except between SabCG, Dichloran-18% glycerol (DG18) and V8® Original juice agar media which showed a significant difference. Also concluded that SabCG can be used as a standard medium. The manuscript is well written.

For the most part, this manuscript can provide relevant results in the fungal biology field. However, this manuscript should need to be revised before it is published.

- Although the journal PLOS ONE does not have a strict limit about paper length, your paper needs to be shortened. The paragraph is too long, some data are redundant, some sentences are too long and confusing. I suggest shortening and reviewing the full paragraph.

WDB: I much appreciate Reviewer #1’s comments and have edited for brevity and clarity where possible. 

- The figures are not clear. Some pictures edges are missing, I suggest you present your pictures in appropriate way to make it clearer for readers.

WDB: I originally used the preflight website system to process all images as per instructions for authors. I cannot seem to replicate the problems described such as missing edges and lack of clarity. I would like further information to help solve these problems. I am using an Apple MacBook Pro / OS X 10.14.6. Perhaps I could correspond directly with the typesetter and use another file format? 

 

Reviewer #2: - First, I would like to thank the author for the manuscript entitled “A comparison of several media types and basic techniques used to assess outdoor Airborne fungi in Melbourne, Australia”

WDB: I very much appreciate Reviewer #2’s very kind words!

- The author tries to compare between widely used media in the detection of airborne fungi, the paper suggests SabCG media as the most detector media for such kind of fungi. The author compares between theses known media and conduct his experiments with different stages and seasons with his own goal for each experiment. He tries to find if any additives or enrichments may increase the ability of detection, identification, and enumeration of airborne fungi, and also he compare between media which can detect xerophilic fungi at low water activity and SabCG media, also he test the CFU detection, identification, and enumeration in case of bacterial contamination, in addition to other experiments.

- Overall I think the research paper needs more organization, add / update some references, clarification of some statistical analysis results, and the following are my comments on it:

In the introduction part,

- Lines 33-36 please add a reference for the information related to acceptable limits of airborne fungi (normal/ maximum). You said there's a lack of widely-accepted limits for maximum permissible and/or normal exposure to occupants, or even what may constitute a ‘moldy’ house. And I think you can find such information’s, Please see this review https://doi.org/10.1080/10473289.1996.10467526 may it will help.

WDB: I thank you for your suggestions and will incorporate them where possible. I should have said ‘universally accepted’ rather than ‘widely’ as this is more correct technically. Strange I did not add this valuable review reference originally, and it has been added now.

In the same paragraph line 34, I have a question, please? What is the difference between identifying fungi within buildings and other places? If the difference exists please mention it, I think you are right with the point of no widely accepted methods for detection and enumeration since no one exactly knows the best method and if you change the method you will get different results. But at least for fungi-identifying, i think we all following the same protocols.

WDB: This is a good question! I shall try to clarify what I originally meant, being the quite broad range of ascomycetes, zygomycetes, yeasts and some basidiomycetes that grow in human-constructed structures such as houses, offices, factories, vehicles, shipping containers, and using the outdoor environment as a base-line defining zero relative-risk (but not zero risk as such). 

This is opposed to food spoilage fungi, plant pathogens, human and animal pathogens, fungi used for other purposes such as composting, fermentation, sewage treatment, fungi collected or cultivated for use as food, or studies of fungi in the natural environment such as leaflitter degradation in forests, etc.

When I started in this field I had attempted to identify fungi recovered from buildings down to species level or similar, but found it difficult, uncertain, and definitely more resource-expensive than what clients were prepared to pay (and beyond what other workers were offering in their services), and ultimately largely unnecessary in the context of determining whether a building has an ecology unlike outdoors and/or more similar to a known water-damaged building. 

I was hoping that by validating a simple, rapid and widely-available test method / medium, then different workers in the same field could more meaningfully compare results and in the future identify any specific organisms more associated with notable disease, if any. That this medium, SabCG, seems able to cultivate a wide range of organisms compared to other common media tested seems beneficial as it may facilitate detection and enumeration. That SabCG seems to allow moulds to grow, develop their morphologies and colours and sporulate fairly consistently helps identification by reference to various ID manuals with their descriptions and images. 

Further specific identification by various means including sub-culture on selective or other special media such as Czapek-Dox, or DNA sequencing is possible but beyond the time and cost constraints imposed by end-users/clients, and the essential question being whether or not the building is of ‘normal’ ecology similar to outdoors and/or a non-water-damaged building. 

I figure that it is best to use whichever medium is found to be the most sensitive and wide-ranging, and most likely to be consistent by curtesy of being simple and closest to being fully ‘defined.’ At least SabCG seems a good compromise for this particular purpose regarding buildings / outdoor environment.

Perhaps I misunderstood the very good and polite question, but will try to clarify what I meant within the manuscript. 

Finally in this paragraph, and about the relation between damped houses and molds, the Ref. 32 in your paper defines or set the correlation between molds and damped houses as you mentioned later.

WDB: The Baudisch et al., 2009 paper (ref #32) used a sieving method to enrich <63 �m effective diameter particles, which they then diluted in saline/detergent with shaking for 30 minutes, and cultured on DG18 and MEA for up to 10 days. Two other airborne dust collection methods were also used for comparison, being a 400-hole system presumably similar to the Andersen 1958 system, and a slit sampler for total airborne counts. They include graphs of their data essentially correlating known mouldy houses with having high viable counts by their method AND high total counts by the slit sampler (hence a high viable : total ratio). Also having a preponderance of Penicillium (or Aspergillus/Eurotium) relative to other organisms (hence an aberrant ecology).

While encouraging, and seemingly designed to overcome problems with samples collected in bulk along with non-fungal materials presumably from carpets via vacuum cleaner and sent by post to the test laboratory, their method may be improved by the use of SabCG rather than DG18 and MEA for reasons shown in my manuscript. 

Further, my professional use of the 400-hole sampler system to sample air (rather than carpets) within the building under assessment seems to avoid the noted problem of significant numbers of >63 �m [um, as micro symbol occasionally doesn’t appear in the .pdf] particles, and avoids the possible issues, expense and time sieving samples, and incubating them for 30 mins in liquid, which may or may not cause some organisms to increase in number by replication, or reduce others’ viability by prematurely germinating under adverse conditions, given it does not seem to have been controlled for in their experiments. 

I noticed a reduction in viability when samples are made wet with bacterial transport gel or similar saline for some time, and hence I collect, store and transport samples dry if they cannot immediately be put on suitable growth media. 

Indeed, the 400-hole sampler top plates I have do occasionally get clogged with particles ~125 µm [um], the diameter of the holes if the site is especially dusty. An ad-hoc in-field experiment was conducted once, attempting to draw dusts out of a carpet suspected of being mouldy. This approach was abandoned when several holes clogged up and proved to be very difficult to clear even with ultrasonic water baths and physical reaming with thin steel wire.

I do like their efforts to clarify and differentiate between a “mouldy” house vs a “dirty” house. This is of interest especially when some workers use LASER airborne total particle counters and attempt to determine if the house/building is mouldy when the counter cannot differentiate between mould and other non-mould dusts, or viable spores vs non-viable spores. I would have liked to have seen the viable : total-count ratio in the Baudisch et al. paper, and further, an attempt to determine if insects were present such as dust mites, carpet beetles, which seem associated with water damaged buildings, mould/fungi and allergy. 

It is my working hypothesis that the symptoms ascribed to being in a mouldy/damp house are more associated with aberrant ecology, especially significant numbers of viable spores of a limited range of types/species/serotypes, thus eliciting a stronger immune response and hence inflammation / allergy. Further, when it comes to a dispute as to whether a house is indeed ‘mouldy’ from a weather or other insurable event rather than merely ‘dirty’ including old mould spores of a wide range of organisms blown into a house over years, this becomes important. 

 

- Line 57-58, fungal growth on building materials always occurs as a result of a high water activity (aw) at the surface of the material for most types of fungi, The minimal water activity for supporting fungal growth was examined by several authors indicating aw-min. values around 0.67–0.75, but growth also depends on temperature, time and the material .and fungus also needs water for 2nd metabolism and the production of mycotoxins which depends on the substrate, aw, temperature, and other environmental factors. Please see Nielsen KF et al. (1999). Production of mycotoxins on artificially and naturally infested building materials. Mycopathologia, 145:43–56.

WDB: At the moment I am merely attempting to which of the media is a reasonable ideal/compromise in estimating viable mould. The question of mycotoxins is very interesting and perhaps there is a correlation between mycotoxin exposure, intoxication, damp houses/buildings and noted symptoms such as rhinitis, but is fraught by the problems of being quite costly, slow turnaround, and of uncertain relationship with food-borne mycotoxins relative to presumably exposure to airborne mycotoxins likely to be an order of magnitude less. 

The idea of the viable-count test is merely to estimate general mouldiness based on common ‘indicator’ organisms found in elevated numbers in damp/mouldy houses/buildings, etc., rather than to find a specific pathogen or mycotoxin. This is similar to the tests for indicator bacteria like E. coli for faecal contamination rather than any particular pathogenic bacteria organisms. 

Further, while it has been put forward that some xerophilic fungi do grow only at low aw, these do not seem to dominate in the experiments performed comparing the low aw medium DG18 with high aw media given the higher numbers and wider range of organisms in high aw media. 

Future studies are envisioned using a single general purpose medium such as SabCG to better study any correlation between viable and total-count mould/yeasts/other fungi, and mycotoxins, and ultimately to determine if there is a link to ‘sick building syndrome’ symptoms. This will be challenging.

Before this, however, there is the more urgent problem of determining if a building such as an inhabited house is mouldy and does require remediation, and whether this remediation has effectively removed mould, rot, weakened timbers, moisture, etc., and presumably mycotoxins and allergenic materials, insects and so-forth. 

- Line 173, (c.f.) refers to what? Please write the word/s from which the abbreviation was made when write it for the first time in the text.

WDB: Many apologies, this is a typo of the commonly used “cf.”, from the Latin “confer”, shorthand meaning ‘to compare.’ I will replace it with ‘compared with’ or similar.

 

- In materials and methods part,

- General comments:

- I think it’s better to add more details for all media you have been used, plus the mixed ones with/out supplements or enrichment/s, all should be included in details, e.g., (glucose-Maltose-peptone media and maltose peptone (please more clarifying in the text ). Details of adding supplements and media enrichments should be presented more clearly in this section instead of the results section later.

WDB: Fixed. 

Line 303, (2-5 knots) please, use SI units or put the equivalent SI unit wit it.

WDB: Fixed.

- Add the method you used to identify the detected fungi to your methods.

WDB: Fixed.

- Results part:

- Line 336, please, where is malt extract media in your results and also in figure (1)? You tested here the maltose utilization in three Maltose based media and also peptone based media. I think the first sentence should be changed to be “Testing various peptone and maltose-based media (Fig 1) in late-January 2020 (summer) in Melbourne, Australia” If I am not mistaken, your results not only showing the maltose utilization but also peptone utilization.

WDB: I have made several significant changes to help clarify. 

- Please add all the formulas of the media you have test in Materials and methods.

WDB: Fixed, and made some inferences in the text later.

 

- Line 356, figure (1).I think it’s good to show your statistical analysis data for variation between and within groups in one table in text, not only in supplementary tables.

WDB: Have added a table.

- In supplementary Table S1.1 (Fig 1): the sample is the total numbers of Identified fungi by collecting a 150 liters sample of air, and CFU/m3 calculated by the equation 1.25( samples detected from 150 L/0.15) but not all CFU/m3 numbers were calculated well, for example, the replicate 2 in media 3 detect 54 fungi with 150L so the CFU/m3 should be 450 but you complete your analysis with 500 CFU/m3 ??? Could you please clarify this? And also why is the big variation in detection within the group, What you think?

WDB: I have fixed the methods section to be more clear about the calculation method according to Andersen, 1958, who presented a table that compensates for the probability of more than one CFU passing through the same hole of the 400-hole plate, and thus appear as one CFU. 

This effect is minimal at low CFU/m^3, but becomes increasingly significant as CFU/m^3 increases. This is akin to the calculation of LD50 and TCID50, and I am told relates to binomial theorem. 

Further, the original Andersen 1958 paper includes a 1.25x factor for use with dried compressed air and plastic Petri dishes and noted problems of static electricity charges and the particles sticking to the plastic. I suspect this is not actually the case with normal humid atmospheric air and hope to address this in a subsequent paper.

I have added a new sheet to the supplementary data Excel spreadsheet. It is based on the Andersen 1958 Table 1, but adjusted to easily allow conversion of CFU/plate from 150 L air, to CFU/m^3 after the 1.25x factor, and is what I use in the laboratory. 

Yes there is significant variation between replicate samples of the same medium. It seems this is a natural phenomenon of the sampling method, slight variations in wind direction, strength, activities in the nearby area, and the normal fungal ecology of the outdoor environment.

In the near future I hope to present another series of experiments in a forthcoming manuscript to test the hypothesis that sampling 150 L over 5 minutes (or 150 L over 10 mins, 60 L over 2 or 4 mins, etc.) yields more consistent results. Also to answer the question whether sampling at 1.5 m or the ground (0 m) has any significant effect, and similarly a variety of wind speeds, etc.

This manuscript/experiment has been held off until a reasonable medium can be established, thus limiting the vast number of replicates required if testing many different media, instead focusing only on one medium such as SabCG.

Many years ago I noticed that when using LASER total air particle counters in sterile rooms / pharmaceutical suites there can be significant transient ‘spikes’ in counts due to the movement and actions of personnel nearby, or drafts and breezes picking up dusts. 

Line 358, the author said (The results suggested that maltose was not commonly utilized by the fungi sampled from the outdoor air, being similar to Peptone CG despite a noted degree of hydrolysis of maltose to glucose) could you please show similarity relation between both?

WDB: I have updated the text to include some estimates of N and C contents of media used, and the amount of glucose in MEA, etc. 

MaltoseCG yielded very few colonies, which were very under-developed and would not normally be counted as true CFU, being very difficult to see, and similar to AgarCG. Other experiments included GlucoseCG [Figure 8] that indicated significantly lower numbers of CFU/m^3 compared with media with peptone and 0-8% glucose, and the GlucoseCG colonies were similar to those on MaltoseCG, being significantly under-developed.

In Figure 1, however, the similar numbers of CFU/m^3 for PeptoneCG (698) and MaltosePeptoneCG (587) suggests that the maltose reduced the numbers of detected CFU, which is odd. 

The similar numbers for MaltoseGlucosePeptoneCG was also unexpected, but as per lines 365-368, possibly an effect of catabolite repression or increased osmolarity, yet this is inconsistent with other media’s 0-8% glucose content, and hence perhaps more likely some form of catabolite repression effect.

Regardless, using GlucosePeptoneCG (=SabCG) and avoiding MEA / malt extract / maltose seems to work better.

Also, pure maltose is indeed hydrolysed during autoclave sterilisation of media into glucose, which is not controlled for or well known, and is indeed present in abundance in MEA (line 371) yet the concentration is not clearly stated nor tested for in standard MEA formulations, and hence I suggest it is preferable for media to have a known amount of glucose instead when counting airborne fungi if only to be more consistent and controlled. 

Also, it seems the standard MEA formula has less Nitrogen content than SabCG.

I suspect that in the past MEA worked reasonably well to culture food spoilage organisms, but is fraught by low nitrogen, batch-to-batch and regional variations in N and maltose/glucose. 

I prefer to keep things as simple as possible, and no simpler. 

Thus on the basis of evidence presented I suggest the best medium and components tested were glucose, the better defined Peptone, and antibiotics, and hence SabCG. 

Perhaps in the future it may be found best to use defined media such as purified essential amino acids, but this is not the greatest, most urgent problem facing this field.

 

- Line 361, I think both carbohydrates (C) source” and (N) both are very important for fungal growth, good sources like peptone will cause excessive growth with knowing that, the C:N ratio is dependent to each fungus type. DOI: 10.1016/j.mycres.2006.07.019.

WDB: Yes. I have added some estimates of this in the text.

The intention is, however, to find a reasonable compromise that is able to cultivate a wide range of organisms reliably, quickly, and cost effectively in the one medium. It will not be perfect, but it seems better than other commonly used media. 

Additionally, the test is not ‘exhaustive,’ being instead more like an indicator such as E.coli for faecal contamination. 

Penicillium, Aspergillus, Cladosporium, etc., seem to grow rapidly in damp buildings and are reasonably quick, cheap and easy to detect, but may or may not actually cause the symptoms reported of people in damp houses, and hence are useful as indicators at least. 

I appreciate the reference given (Li Gao, et al., 2007) and their finding of specific C contents between 0.6% - 1.2% (6 - 12 g/L, 10-80:1 C:N) was the best for each of their specific fungal organisms under test. 

In a way I performed a form of this type of C : N comparison and optimisation in one of the experiments in which a range of glucose concentrations were compared from 0 – 8% (0 – 80 g/L). It seemed there was a vague maximum of numbers of CFU detected in outdoor air at about 4% (40 g/L). The organisms Li Gao, et al., used seem specific to biological control and hence selected and sub-cultured for a specific purpose, whereas the organisms found outdoors are wild-type, and for a different purpose. Using SabCG I do occasionally find Paecilomyces, and also Trichoderma species, the latter only in heavily mouldy and wet houses, presumably because it relies on the presence of other fungi for its growth.

- Line 387, no statistical information

WDB: Added new table, Table 1.

- In supplementary Table S1.2 (Figs 2 and 3): please add the significance level and statistical analysis.

WDB: Added.

- Line 396, Fig 3, the reflection of light on the image affects the quality and 1st Replicate of celery juice (cooked) ??? very low detection rate with no any comparison with other replicates???

WDB: Sadly I did not think I’d include these images when I took them! They are only a small number of the total number of plates. They are for illustrative purposes only. I do have images of each individual plate, but these also are just rough lab documentation images with the general lab bench and in-use items visible in the background, including the microscope. 

The data was actually in the counting and ID of colonies on the physical plates at the time, not from images. I find it more reliable / easier / quicker. The images are just for reference. The images presented below were from somewhat after 3 days incubation when actual counting/ID was done, and hence some satellite / progeny colonies are occasionally seen but had no effect on counts. This was done to check the ID of some slower growing or otherwise smaller pale colonies.

I could simply delete the sections of images with reflections if you’d like.

Later images were taken with overhead lights turned off to prevent this issue, but did make the colonies appear more like silhouettes and hence without colouring often needed for ID. I have since set up a single light source that can be positioned in a manner that casts oblique light, avoiding bright reflections.

- Line 406-408: it’s better to talk about this later, after the comparison between V8c, DG18c and SabCG media or in discussion, with hence that the conditions were not the same.

WDB: Fixed. 

- Line 412, please use SI units.

WDB: Fixed

- Line 418-420 Fig4, please show your statistical analysis in one table.

WDB: Fixed (Table 1)

- Fig 5, (a) vc8, (b) DG18,(c) Sab. By checking the images, I noticed that the number of colonies in vc8 media which was mentioned as a poorly detector is higher among others?

WDB: As above, these are just a small subset of the complete set of plates, and the images are not perfect, having features that often look like colonies but are perhaps digital artefacts, droplets of condensate on lids, and in the V8 media, lumps of starch and other materials even prior to incubation, and other things that stain darkly on addition of Lugol’s Iodine, etc. 

The Lugol’s Iodine effect is short-lived, lasting only about 15-20 minutes, fading noticeably during that time from rich dark red to a mottled pale red-brown.

As above, I do have individual images and counted/ID’d from the original plates at 3 days rather than images.

- Line 542, Flame sterilization/sanitization of 400-hole Andersen air Sampling top-plate, please add the details in material and methods.

WDB: Added/fixed.

 

- Finally, I think more organizing of the manuscript, doing the comparison between Sab, DG18, V8, MEA, PDA, etc., in the same conditions and different seasons with more replicates , because variations within each media are high, and showing the new results data with other experiments that author already conduct will be useful.

WDB: 

I was hoping that the data demonstrated that at very least that complicated media with mineral supplements, vegetable supplements, low aw, and the presence of starch is less ideal than a simple general purpose medium to estimate fungi outdoors/indoors. 

A simple medium like SabCG is inherently less subject to batch to batch variation, or variation between different workers in different regions. 

I intend to use SabCG for further investigations as you suggest to help simplify and streamline them, using more replicate plates, likely 6-10x. Perhaps comparing DG18CG, SabCG indoors/outdoors, just to check if there is a shift in ecology in some buildings known to be WDB. Also the questions of air velocity, sampler height, humidity, etc.

I also intend validating the Andersen 400-hole sampler itself. I suspect that the use of the 1.25x modifier stipulated by Andersen, 1958, was intended for use with plastic Petri dishes and very dry compressed air rather than normal humid atmosphere. 

Much work is required to refine and validate the basic tools used by myself and fellow Indoor Environmental Professionals.

---

## [Decision Letter · Decision Letter 1]

16 Nov 2020

PONE-D-20-26205R1

A comparison of several media types and basic techniques used to assess outdoor airborne fungi in Melbourne, Australia.

PLOS ONE

Dear Dr. Black,

Thank you for submitting your manuscript to PLOS ONE. After careful consideration, we feel that it has merit but does not fully meet PLOS ONE’s publication criteria as it currently stands. Therefore, we invite you to submit a revised version of the manuscript that addresses the points raised during the review process.

We look forward to receiving your revised manuscript.

Kind regards,

Zonghua Wang, Ph.D.

Academic Editor

PLOS ONE

Reviewers' comments:

Reviewer's Responses to Questions

**Comments to the Author**

1. If the authors have adequately addressed your comments raised in a previous round of review and you feel that this manuscript is now acceptable for publication, you may indicate that here to bypass the “Comments to the Author” section, enter your conflict of interest statement in the “Confidential to Editor” section, and submit your "Accept" recommendation.

Reviewer #1: All comments have been addressed

Reviewer #2: All comments have been addressed

2. Is the manuscript technically sound, and do the data support the conclusions?

Reviewer #1: Yes

Reviewer #2: Yes

3. Has the statistical analysis been performed appropriately and rigorously? 

Reviewer #1: Yes

Reviewer #2: Yes

4. Have the authors made all data underlying the findings in their manuscript fully available?

Reviewer #1: Yes

Reviewer #2: Yes

5. Is the manuscript presented in an intelligible fashion and written in standard English?

Reviewer #1: Yes

Reviewer #2: Yes

6. Review Comments to the Author

Reviewer #1: Overall the author did a good job addressing my comments.

There is still only one issue that need to be addressed:

Introduction: I think the author can make the introduction even shorter; I see your need in outlying the framework and prior research; but I think that in order to attract the reader to read the paper, you want to get to your main point earlier; I do not have specific suggestions how to do it; and you don’t have to agree with this suggestion, of course; it is just a suggestion to enhance readability of the paper.

Reviewer #2: I thank the author again, the author answered most of my concerns about the MS, traying his best to make the MS more clear and acceptable, I was concern about the conditions of the experiments conducted like small number of replicates and samples size which cause high variation in results within the same media. I think he has made an expressed work in his MS. All the materials and methods are clear now, deleting some photos is possible but at the same time providing good photos for the key experiments will be much appreciated for publication, any possible improvement with more investigations as you indicated in your answers will be also highly appreciated and add much value to this paper.

7. PLOS authors have the option to publish the peer review history of their article (what does this mean?). If published, this will include your full peer review and any attached files.

Reviewer #1: No

Reviewer #2: No

---

## [Author Response · Author response to Decision Letter 1]

24 Nov 2020

Response to Reviewers

WD Black (WDB), 17-25 Nov 2020

Noted: all other sections both reviewers affirm compliance/acceptance, except Section 6 in which this is either unclear or noted as a helpful suggestion. 

I thank the reviewers for their comments and attempt to address them below.

***

6. Review Comments to the Author

Reviewer #1: Overall the author did a good job addressing my comments.

There is still only one issue that need to be addressed:

Introduction: I think the author can make the introduction even shorter; I see your need in outlying the framework and prior research; but I think that in order to attract the reader to read the paper, you want to get to your main point earlier; I do not have specific suggestions how to do it; and you don’t have to agree with this suggestion, of course; it is just a suggestion to enhance readability of the paper.

I thank Reviewer #1 for their suggestions. 

I have attempted to address these by a significant ‘haircut’ of the introduction to weed out material I realise was written with a non-scientific audience in mind. I realise they’d be unlikely to read the paper in any guise, and can better make my point to them by other means such as PowerPoint and merely referring to the paper. 

The main concern I have is that I will be criticised for not thoroughly reviewing the literature, and also not framing the problems I attempt to address in the series of experiments I performed. If you see this is OK as-is now, then all good. 

There are of course a number of questions asked of me by that non-scientific audience I was hoping to answer in this more formal scientific forum by way of the longer introduction. I think I have achieved a balance here in cutting it down to 1800 words or so, trimming out 80 lines. 

I’ve repositioned some sentences to better streamline the reading flow. 

 

Reviewer #2: I thank the author again, the author answered most of my concerns about the MS, traying his best to make the MS more clear and acceptable, I was concern about the conditions of the experiments conducted like small number of replicates and samples size which cause high variation in results within the same media. I think he has made an expressed work in his MS. All the materials and methods are clear now, deleting some photos is possible but at the same time providing good photos for the key experiments will be much appreciated for publication, any possible improvement with more investigations as you indicated in your answers will be also highly appreciated and add much value to this paper.

I again thank Reviewer #2 for their valuable comments. 

To explain, I found it difficult to compare many different media on the same day because of how quickly and dramatically Melbourne weather changes. It is famous for this. So I was obliged to limit the total number of plates per day to best have everything as similar as possible, highlighting any differences in media rather than weather. 

I have tried to not burden the readers with too much statistical data, instead using graphs that hopefully tell the story in a readily accessible way, especially the non-scientific readers I am expecting. 

In the revised latest manuscript (attached) some p-values are on lines 283-285 (p = 0.0089, three replicate plates), and line 367 (p < 0.0003, six replicate plates), suggesting that the null-hypothesis (H0) of there being no real difference between these media tested, being merely random chance is less than 1 in 113 and 1 in 3333, respectively. Obviously this is better than the often cited p < 0.05 limit, being 1 in 20. 

I am therefore fairly confident that these experiments have demonstrated a significant difference between different media as described/presented even at three replicate plate numbers. 

I hope I have shown that some of the media are clearly not suitable for this purpose even with three replicate plates only, and logically supported by the evidence that not all random wild airborne fungi are able to digest starch and/or maltose, and hence the use of PDA, V8, MEA is not supported. 

I have added the calculated P-value for Fig 2 (lines 322-323), and a comment on line 412.

A quick check of my data indicates that when one takes the six SabCG replicates in groups of three in every combination, the SDs/means vary around the mean by 36%, whereas the full set of six varies by 29%, a difference of 7%. As the number of replicate samples increases, the difference becomes less, but asymptotically, approaching zero but never reaching it. 

I suppose when there is a large difference in the media via the number of CFU, one needs fewer replicate plates to see this difference. This is also visible in the figures wherein the +/-SD bars often do not overlap.

Some data is not really able to be meaningfully compared by these means, being more qualitative such as the media clouding (the two mineral supplemented media) or colonies being strangely shaped and under-developed when bacteria are present in number (antibiotic-free media).. 

As mentioned, I hope to move beyond investigating the media, and into investigating the differences between mouldy and non-mouldy houses, and the accuracy/precision of test methods using the best candidate medium, SabCG. I will be using more replicates where possible, of course. 

I noticed that the quality of the images embedded in the .pdf sent to reviewers are not at the high quality of the original images. I suspect that the final version will be of high enough quality for the readers to see beyond the light reflections, and I hope they also understand these images are really just ‘examples’ rather than the actual data. A little embarrassing for me I admit. I was more focused on the colonies rather than the image itself.

---

## [Editor Report · Decision Letter 2]

26 Nov 2020

A comparison of several media types and basic techniques used to assess outdoor airborne fungi in Melbourne, Australia.

PONE-D-20-26205R2

Dear Dr. Black,

We’re pleased to inform you that your manuscript has been judged scientifically suitable for publication and will be formally accepted for publication once it meets all outstanding technical requirements.

Kind regards,

Zonghua Wang, Ph.D.

Academic Editor

PLOS ONE
---

## [Editor Report · Acceptance letter]

3 Dec 2020

PONE-D-20-26205R2 

A comparison of several media types and basic techniques used to assess outdoor airborne fungi in Melbourne, Australia 

Dear Dr. Black:

I'm pleased to inform you that your manuscript has been deemed suitable for publication in PLOS ONE. Congratulations! Your manuscript is now with our production department. 

Kind regards, 

on behalf of

Prof. Zonghua Wang 

Academic Editor

PLOS ONE